# Sucrose-driven carbon redox rebalancing eliminates the Crabtree effect and boosts energy metabolism in yeast

Zhiqiang Xiao [1,2,3], Yifei Zhao[1,2,3], Yongtong Wang[1,2,3], Xinjia Tan[1,2,3], Lian Wang[4], Jiwei Mao[5], Siqi Zhang[1,2,3], Qiyuan Lu[1,2,3], Fanglin Hu[1,2,3], Shasha Zuo[1,2,3], Juan Liu [2,3] ✉ & Yang Shan [1,2,3] ✉

*Saccharomyces cerevisiae* primarily generates energy through glycolysis and respiration. However, the manifestation of the Crabtree effect results in substantial carbon loss and energy inefficiency, which significantly diminishes product yield and escalates substrate costs in microbial cell factories. To address this challenge, we introduce the sucrose phosphorolysis pathway and delete the phosphoglucose isomerase gene *PGI1*, effectively decoupling glycolysis from respiration and facilitating the metabolic transition of yeast to a Crabtree-negative state. Additionally, a synthetic energy system is engineered to regulate the NADH/NAD⁺ ratio, ensuring sufficient ATP supply and maintaining redox balance for optimal growth. The reprogrammed yeast strain exhibits significantly higher yields of various non-ethanol compounds, with lactic acid and 3-hydroxypropionic acid production increasing by 8- to 11-fold comparing to the conventional Crabtree-positive strain. This study describes an approach for overcoming the Crabtree effect in yeast, substantially improving energy metabolism, carbon recovery, and product yields.

Growing concerns over environmental friendliness and resource sustainability have driven interest in the bioproduction of a wide range of chemicals and food substrates[1]. However, to make bioproduction viable on a large scale, the optimisation of carbon utilization is essential. This optimisation is crucial for achieving carbon neutrality, which remains the foremost challenge in microbial biosynthesis[2,3]. Achieving this goal requires maximising output while minimising raw material input. Rewiring the central carbon metabolism of microorganisms is a key strategy for enhancing carbon source utilization. However, microbial metabolism involves numerous interdependent chemical reactions that form a complex and dynamically balanced network[4,5]. Due to the highly complex and interconnected topological structure of metabolic networks, altering a single reaction can trigger a wide cascade of effects, complicating the adjustment and modification of these networks[6–8]. Many of these reactions are vital for core cell function and active metabolism, making it challenging to adjust metabolic networks without disrupting normal cellular activities[9]. Therefore, the central challenge lies in maximising carbon utilization while preserving cell vitality.

*Saccharomyces cerevisiae* is widely used as a model organism in industrial production due to its well-characterised genetic background and strong robustness[10]. However, it prioritises alcoholic fermentation over respiration for energy production when glucose is abundant[11], a phenomenon known as the Crabtree effect. In *S. cerevisiae*, one glucose molecule can yield 32 molecules of ATP through respiration, whereas fermentation produces only two molecules of ATP[12]. The low

[1]Longping Agricultural College, Hunan University, Changsha 410125, China. [2]Hunan Institute of Agricultural Product Processing and Quality Safety, DongTing Laboratory, Hunan Academy of Agricultural Sciences, Changsha 410125, China. [3]Hunan Key Lab of Fruits & Vegetables Storage, Processing, Quality and Safety, Hunan Agricultural Products Processing Institute, Changsha 410125, China. [4]Frontier Science Center for Synthetic Biology (Ministry of Education), Key Laboratory of Systems Bioengineering, and School of Chemical Engineering and Technology, Tianjin University, Tianjin 300072, China. [5]Department of Life Sciences, Chalmers University of Technology, SE412 96 Gothenburg, Sweden. ✉e-mail: liujmax2019@163.com; sy6302@sohu.com

energy efficiency of the fermentation pathway limits biomass production and the synthesis of non-ethanol chemicals, particularly when abundant redox power is required[13,14]. The metabolic shift induced by the Crabtree effect is generally attributed to the higher enzymatic efficiency of glycolysis[15]. The ability to synthesise ATP per unit mass is significantly higher for glycolytic enzymes than for respiratory enzymes[16]. Additionally, the high glycolytic flux suppresses

respiration-related genes[17], causing pyruvate to be rapidly converted to ethanol rather than entering the mitochondria[18]. Several strategies have been explored to engineer Crabtree-negative strains, each with distinct advantages and limitations (Supplementary Data 1). For instance, engineering Crabtree-negative yeast involves deleting the genes *PDC1/5/6*, which encode pyruvate decarboxylase, thereby preventing ethanol accumulation[13]. However, this approach leads to

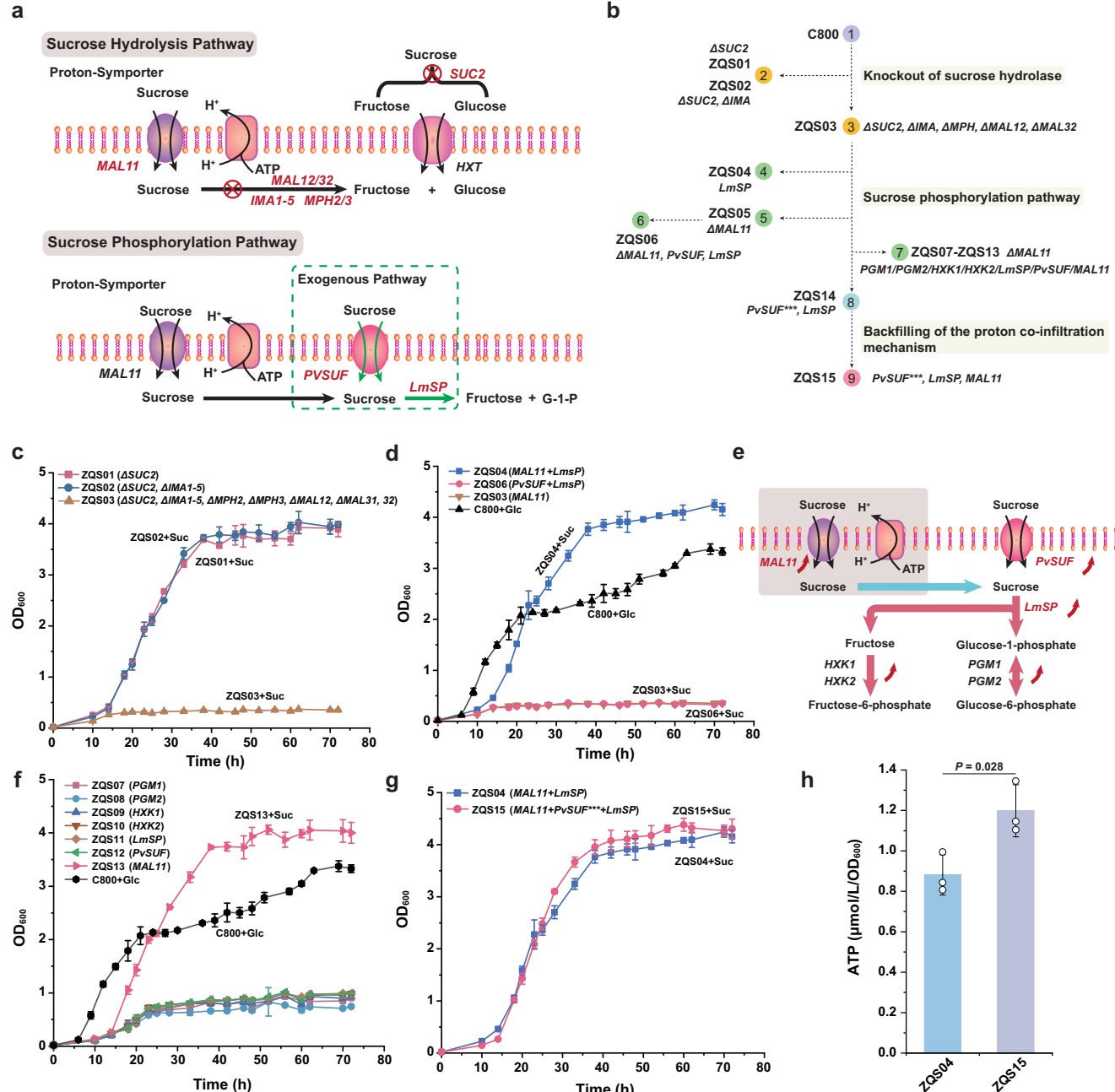

**Fig. 1 | Design and construction of the sucrose phosphorylation pathway.**
**a** Replacement of the endogenous hydrolysis pathway with the sucrose phosphorylation pathway. **b** Flowchart of strain construction using the CRISPR/Cas9 system. Different strains were colour-coded into three main parts: knockout of sucrose hydrolase, introduction of the sucrose phosphorolysis pathway, and backfilling of the proton co-infiltration mechanism. **c** Growth of strains with knocked-out genes related to the sucrose hydrolysis pathway when cultured in YPD medium with sucrose as the sole carbon source. **d** Growth of strains containing sucrose phosphorylase LmSP and the exogenous transporter PvSUF in sucrose medium. The growth curve of strain C800 in a glucose medium serves as a control.

**e** Schematic diagram of overexpressed key genes in the sucrose phosphorolysis pathway. **f** Growth of strains with overexpressed genes in the sucrose phosphorylase pathway in sucrose medium, with C800 as the control strain. **g** Growth of strains co-expressing the mutant *PvSUF*[209F C26SF G326C] and *MAL11* dual transport proteins, with strains containing only *MAL11* and *LmSP* as the control.
**h** Comparison of ATP consumption between the dual transport protein strain ZQS15 and the strain ZQS04 containing MAL11. All data are expressed as mean ± SD of biological triplicates. Statistical analysis was conducted using the Student's *t*-test (two-tailed; sample size, *n* = 3). Source data are provided as a Source Data file.

insufficient acetyl-CoA supply and inadequate $NAD^+$ regeneration, leading to an imbalance in the $NADH/NAD^+$ ratio, which impairs cell growth[19]. Pyruvate decarboxylase-negative strains cannot grow in glucose-rich media[19]. An alternative approach for overcoming the Crabtree effect is to increase NADH oxidation for ATP synthesis. However, this approach cannot completely eliminate overflow metabolism, leading to carbon wastage[20]. Therefore, we believe that the principal strategy to overcoming the Crabtree effect is to rewire central carbon metabolism without disrupting the $NADH/NAD^+$ balance. To achieve this goal, it is necessary to adjust the relationship between glycolytic flux and the tricarboxylic acid (TCA) cycle while ensuring efficient $NAD^+$ regeneration.

Yeast can utilize various carbon sources. First-generation raw materials like sugars and starches come from agricultural production. Second-generation raw materials are agricultural byproducts, such as lignocellulosic sugars extracted from straw or corn stover or glycerol from biodiesel production[21,22]. In industrial applications, converting abundant, inexpensive sucrose from sugarcane or sugar beet is particularly attractive[23,24]. Sucrose is metabolised in two ways: either by extracellular hydrolysis followed by absorption of the resulting fructose and glucose via facilitated diffusion or intracellular hydrolysis after uptake through a proton-symport mechanism[25] (Fig. 1a and Supplementary Fig. 1a). However, yeast cannot store the free energy released during sucrose hydrolysis ($\Delta G_0' = -29$ kJ/mol)[26]. Efficient free-energy conservation impacts product yield[27]. In ATP-dependent biosynthesis, improved energy conservation directs more carbon toward the desired product[15,28]. Certain microorganisms enzymatically break down sucrose into fructose and glucose-1-phosphate, bypassing the ATP requirement for hydrolysis and conserving more free energy (Supplementary Fig. 1b)[29,30]. Moreover, sucrose phosphorolysis effectively mitigates glucose repression and facilitates the entry of both fructose and glucose-1-phosphate into central carbon metabolism[31]. In sucrose metabolism, knocking out the *PGI1* gene can reduce the crosstalk between the Embden-Meyerhof-Parnas (EMP) pathway and the pentose phosphate pathway (PPP). This advancement is crucial for understanding and potentially mitigating the Crabtree effect.

In this work, we transit yeast into a Crabtree-negative state by partitioning carbon metabolism based on sucrose phosphorolysis. It improves the energy metabolism of the strain and redirects flux away from ethanol production. Sucrose phosphorolysis facilitates the use of fructose and glucose-1-phosphate as carbon sources. We address the crosstalk between the EMP pathway and PPP, reducing flux to the former and successfully engineering a Crabtree-negative strain (Supplementary Fig. 1c). Transcriptome sequencing confirms robust respiratory capacity and ATP production in the resulting strain (Supplementary Fig. 1d). To mitigate the prolonged lag phase, we identify and target genes regulating sucrose phosphorolysis, reducing the lag phase by 30.8% (Supplementary Fig. 1e). Subsequently, a synthetic energy system is developed by overexpressing the NADH dehydrogenase genes, *NDE1* and *NDE2*, to channel more NADH into the respiratory chain for ATP synthesis (Supplementary Fig. 1f). The growth of the Crabtree-negative strain with the synthetic energy system on sucrose enhances its synthetic capabilities, resulting in increased production of several industrially relevant products, including lactic acid, 3-hydroxypropionic acid (3-HP), *p*-coumaric acid, and farnesene (Supplementary Fig. 1g).

## Results
### Sucrose phosphorolysis replaces the native hydrolysis route
Sucrose metabolism in yeast is governed by several key genes: the extracellular sucrose-hydrolysing enzyme gene *SUC2*, the maltase genes *MAL12* and *MAL32*, isomaltase genes *IMA1-5*, and α-glucoside permease genes *MPH2* and *MPH3*[25] (Fig. 1a). These enzymes have the potential to hydrolyse sucrose into glucose and fructose. SUC2 catalyzes the extracellular hydrolysis of sucrose, whereas the other genes

facilitate its intracellular hydrolysis. Additionally, the main sucrose transporters in *S. cerevisiae* are MAL11 and MAL31. To ensure the complete phosphorylation of sucrose into fructose and glucose-1-phosphate, we systematically deleted the genes associated with sucrose hydrolysis as well as the sucrose transporter *MAL31* using a multi-gene editing plasmid while retaining *MAL11* to maintain normal sucrose transport. This resulted in the construction of strains ZQS01-ZQS03 (Fig. 1b). When cultivated in YPD medium with sucrose as the sole carbon source, ZQS03 exhibited impaired growth, while ZQS01 and ZQS02 retained their ability to metabolise sucrose (Fig. 1c and Supplementary Fig. 2a). Thus, ZQS03 was selected for further investigation due to its inability to hydrolyse sucrose.

To validate sucrose phosphorolysis, we introduced the *LmSP* gene from *Leuconostoc mesenteroides*[26] into ZQS03, which enables the phosphorolysis of sucrose into fructose and glucose-1-phosphate. As a result, strain ZQS04 restored sucrose metabolism and exhibited a higher maximum specific growth rate ($\mu_{max}$) of $0.20\ h^{-1}$ compared to the parental strain C800 with a $\mu_{max}$ of $0.18\ h^{-1}$ (Fig. 1d and Supplementary Fig. 2b). It indicates that the sucrose phosphorolysis pathway provides a growth advantage over glucose metabolism. In yeast, sucrose transport relies on the endogenous sucrose-proton symporter encoded by *MAL11*, which requires one ATP per transported sucrose molecule[25]. It has been reported that PvSUF, derived from *Phaseolus vulgaris*, can transport sucrose into the cell without consuming ATP[26]. To improve energy efficiency, the symporter was replaced with an assisted diffusion transport system by introducing the *PvSUF* gene. However, the resulting strain ZQS06, in which *MAL11* was replaced with *PvSUF* and *LmSP*, exhibited impaired sucrose utilization (Fig. 1d). Increasing sucrose concentration did not significantly improve growth (Supplementary Fig. 2c), indicating that sucrose transport inefficiency was not due to substrate availability.

To identify the rate-limiting step in the sucrose phosphorolysis pathway, we overexpressed related genes (Fig. 1e). Strain ZQS13 with replenished *MAL11* expression recovered growth in the sucrose medium with a $\mu_{max}$ of $0.20\ h^{-1}$ (Fig. 1f and Supplementary Fig. 2b). This indicates that the sucrose transport capacity of PvSUF alone is inadequate to sustain normal yeast metabolism. To circumvent the transport constraint, we integrated the mutant version, $PvSUF^{I209F\ C265F\ G326C}$ and LmSP into ZQS05, which is believed to exhibit enhanced sucrose transport efficiency compared to PvSUF, allowing the strain to regain growth[32]. The resulting strain ZQS14 regained its sucrose utilization capability but exhibited a prolonged lag phase (Supplementary Fig. 2d). Therefore, we co-expressed the mutated $PvSUF^{I209F\ C265F\ G326C}$ and *MAL11* resulted in strain ZQS15, which achieved a $\mu_{max}$ of $0.21\ h^{-1}$ (Fig. 1g and Supplementary Fig. 2b). Meanwhile, this dual transporter mechanism increased ATP levels (Fig. 1h), suggesting that part of sucrose was transported through the mutant PvSUF without ATP consumption. Consequently, ZQS15 became the base strain for the sucrose phosphorolysis pathway, demonstrating an energy-efficient mechanism for sucrose transport and utilization in yeast.

### Transitioning yeast metabolism to a Crabtree-negative state
The Crabtree effect, in which yeast favours fermentation over respiration in high-glucose environments, leads to ethanol accumulation. This phenomenon primarily results from the accumulation of NADH in the cytosol due to high EMP pathway flux[33], prompting yeast cells to restore the $NADH/NAD^+$ balance via ethanol and glycerol pathways[20]. To assess the impact of sucrose phosphorolysis on the Crabtree effect, we compared ethanol accumulation in strain ZQS15 grown in sucrose versus glucose medium. The results showed that ZQS15 exhibited reduced ethanol accumulation (Fig. 2a) in the presence of sucrose without impairment in growth (Fig. 2b). It is speculated that the slower transport rate of sucrose compared to glucose allows for a more gradual release of sucrose into the cell, potentially mitigating the Crabtree effect. Additionally,

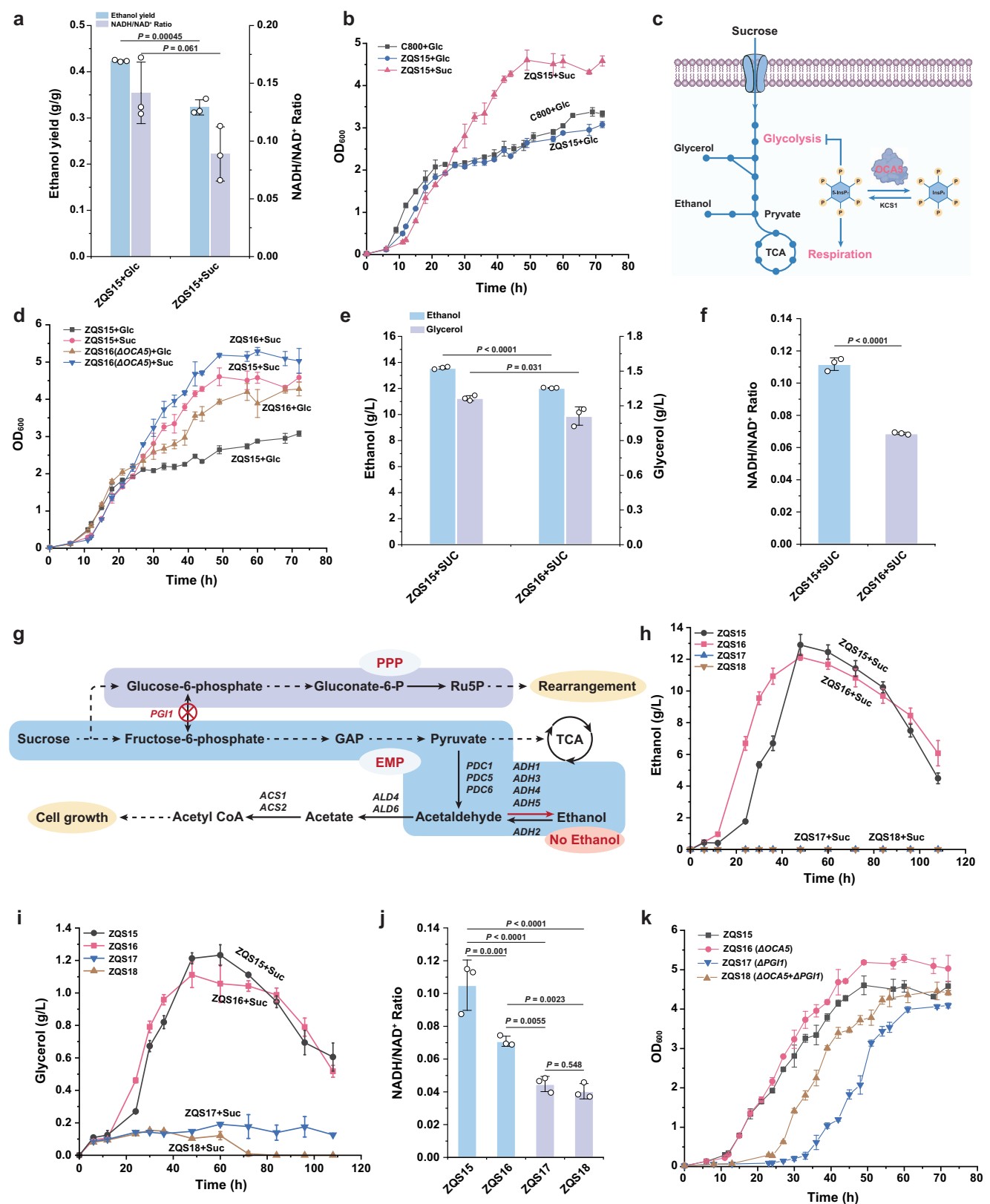

we compared the NADH/NAD$^+$ and NADPH/NADP$^+$ ratios between the two metabolic modes. The reduced NADH/NAD$^+$ ratio observed during sucrose phosphorolysis (Fig. 2a) indicates an improved redox balance, partially shifting intracellular metabolism toward a respiration-dependent state. In contrast, the NADPH/NADP$^+$ ratio remained unchanged (Supplementary Fig. 3a). Given that the pentose phosphate pathway (PPP) is the primary source of NADPH, this suggested that sucrose phosphorolysis does not affect PPP metabolism.

Central carbon metabolism balances flux between the EMP and respiratory pathways, ensuring a continuous supply of energy, cofactors, and essential precursors required for anabolism[2]. Inositol pyrophosphatase (OCA5) plays a pivotal role in regulating the EMP pathway and respiration, and its deletion alleviates glycolysis-induced

**Fig. 2 | Establishing a Crabtree-negative strain based on the sucrose phosphorolysis-capable strain. a** Comparison of ethanol production capacity and NADH/NAD$^+$ ratio between the sucrose phosphorylase and glucose metabolism models. **b** Comparison of growth between the sucrose phosphorolysis and glucose metabolism models. **c** Functional schematic diagram of inositol pyrophosphatase OCA5 regulates glycolysis and respiration by adjusting levels of 5-diphosphoinositol 1,2,3,4,6-pentakisphosphate (5-InsP7). **d** Growth of the *OCA5*Δ strain under different metabolic conditions, with ZQS15 as the control. **e** Accumulation of ethanol and glycerol in the *OCA5*Δ strain under sucrose phosphorolysis metabolism mode. **f** Comparison of the NADH/NAD$^+$ ratio between the *OCA5*Δ strain and the wild-type strain under sucrose phosphorolysis metabolism mode. **g** Schematic diagram of metabolic changes in the *PGI1*Δ strain under sucrose phosphorolysis mode. **h** Ethanol accumulation in *PGI1*Δ strains compared to *PGI1* wild-type strains. ZQS17 and ZQS18 are *PGI1*Δ strains, while ZQS15 and ZQS16 are control strains. **i** Glycerol production in *PGI1*Δ strains compared to *PGI1* wild-type strains. **j** Changes in the *PGI1*Δ strain's NADH/NAD$^+$ ratio. **k** Impact of *PGI1* knockout on the growth cycle of different strains. All data are expressed as mean ± SD of biological triplicates. Statistical analysis was conducted using the Student's *t*-test (two-tailed; sample size, *n* = 3). Source data are provided as a Source Data file.

inhibition of respiration (Fig. 2c)[17]. Consequently, we knocked out the *OCA5* gene in strain ZQS15 to generate strain ZQS16, which exhibited a similar growth profile to ZQS15 (Fig. 2d), with a μ$_{max}$ of 0.21 h$^{-1}$ (Supplementary Fig. 4a). However, ZQS16 demonstrated lower ethanol and glycerol accumulation (Fig. 2e and Supplementary Fig. 3b, c) and a reduced NADH/NAD$^+$ ratio (Fig. 2f and Supplementary Fig. 3d), indicating enhanced respiratory capacity and alleviation of the Crabtree effect. Meanwhile, the NADPH/NADP$^+$ ratio remained unchanged (Supplementary Fig. 3e), suggesting that the deletion of *OCA5* does not affect the PPP.

In the sucrose phosphorolysis model, the *PGI1* gene serves as a crucial link between the EMP pathway and PPP (Fig. 2g), enabling the EMP pathway to maintain a high flux, which results in overflow metabolism and the accumulation of ethanol and glycerol. To validate this hypothesis, we knocked out the *PGI1* gene in strains ZQS15 and ZQS16, generating ZQS17 and ZQS18, respectively. As expected, the *PGI1*Δ strain could not grow in glucose medium due to the absence of glycolysis (Supplementary Fig. 5). However, in the sucrose medium, the EMP pathway is initiated by fructose, bypassing the role of *PGI1*. Examination of Crabtree effect-related parameters in ZQS17 and ZQS18 revealed no ethanol accumulation and minimal glycerol accumulation (Fig. 2h, i), along with a further reduction in the NADH/NAD$^+$ ratio (Fig. 2j). These results indicate that *PGI1*Δ strains exhibit a Crabtree-negative phenotype in the sucrose phosphorolysis pathway. Notably, although the μ$_{max}$ of the *PGI1* knockout strains ZQS17 and ZQS18 were like that of ZQS16 (Supplementary Fig. 4a), both exhibited a longer lag phase time compared to ZQS16 (Supplementary Fig. 4b). We speculate that the knockout of *PGI1* reduces the EMP pathway flux, thereby extending the cell growth cycle. Moreover, we found that the lag phase time of ZQS17 reached 34.1 h, significantly longer than that of ZQS18 with 23.5 h (Supplementary Fig. 4b), indicating that *OCA5* deletion improves the growth performance of Crabtree-negative yeast strains. Additionally, the NADPH/NADP$^+$ ratio in ZQS17 and ZQS18 was lower compared to ZQS15 and ZQS16 (Supplementary Fig. 3f), indicating that the knockout of *PGI1* reduces the PPP flux. This is due to the inability of the EMP to supply glucose-6-phosphate (G6P), a key substrate for the PPP, following *PGI1* deletion. The limitation of G6P results in a reduced NADPH production, thereby lowering the NADPH/NADP$^+$ ratio.

### Transcriptional profiles of the Crabtree-negative strains
To investigate the Crabtree-negative phenotype in *PGI1*Δ strains under sucrose phosphorolysis, we performed RNA-Seq transcriptome analyses on ZQS16 (*OCA5*Δ) and ZQS18 (*OCA5*Δ, *PGI1*Δ). The results revealed significant gene expression differences between ZQS18 and ZQS16 (Supplementary Fig. 6a), with 1,471 differentially expressed genes (DEGs), including 898 upregulated and 573 downregulated genes (Supplementary Fig. 6b). Kyoto Encyclopaedia of Genes and Genomes (KEGG) enrichment analysis showed upregulation of pathways related to carbon metabolism and the TCA cycle (Supplementary Fig. 7a), indicating enhanced oxidative phosphorylation, metabolic biosynthesis, and conversion of carbon sources into metabolites in ZQS18[15]. Conversely, ribosome biogenesis was downregulated (Supplementary Fig. 7b), suggesting reduced cell growth potential, possibly explaining the extended lag phase observed (Fig. 2k)[34]. To further

explore the impact of *PGI1*Δ on central carbon metabolism, we analysed DEGs associated with carbon metabolism. In the ZQS18+Suc group, all genes involved in the EMP pathway were downregulated, while those in the PPP and TCA cycles were upregulated (Fig. 3a). This demonstrates that the absence of *PGI1* restricts EMP flux while increasing flux through the TCA cycle and PPP. Consequently, *PGI1*Δ strains with reduced EMP flux are more inclined to rely on respiration rather than fermentation for energy production.

Gene expression analysis of the electron transport chain (ETC) revealed a significant increase in metabolic flux through the respiratory chain (Fig. 3b). The ETC, comprising Complexes I–IV and ATP synthase (Complex V), is crucial for electron transfer and ATP production[35]. Transcriptome analysis showed a significant increase in the flux of NADH dehydrogenases NDE1/2 and NDI1, which are equivalents of Complex I. This indicates an enhanced capacity for converting NADH to NAD$^+$, with most of the NADH being utilised for ATP synthesis via the ETC, thereby explaining the lower NADH/NAD$^+$ ratio in ZQS18 (Fig. 2j). Additionally, the flux through Complexes II–IV increased significantly, indicating a marked enhancement in ATP generation via oxidative phosphorylation in ZQS18 compared to ZQS16.

### Optimising cell growth under sucrose phosphorolysis mode
Because of the extended lag phase observed in Crabtree-negative strains utilising sucrose metabolism, we analysed transcriptome data to identify endogenous genes in yeast that may regulate cell growth. ZQS16/ZQS15 strains exhibited higher μ$_{max}$ on sucrose than on glucose, while ZQS18/ZQS17 (*PGI1*Δ) showed a longer lag phase time on sucrose than ZQS16/ZQS15 (Supplementary Fig. 8). Based on these observations, the comparative datasets were categorised into groups (Fig. 4a). Comparisons I, II, and III revealed 25 DEGs (Fig. 4b), while Comparisons IV, V, and VI identified 462 DEGs (Fig. 4c). Comparisons I and IV represent differences between the sucrose phosphorolysis metabolism and glucose metabolism, whereas Comparisons III and VI focus on the effect of *PGI1* deletion. Given that strains utilizing sucrose phosphorolysis exhibited a higher μ$_{max}$, while *PGI1* deletion strains experienced a prolonged lag phase time, we hypothesized that specific gene expression patterns might be associated with these growth differences. To pinpoint key regulators, we focused on genes that were upregulated in Comparisons I and IV but downregulated in Comparisons III and VI, selecting 16 candidate genes with potential roles in promoting cell growth (Fig. 4d). Conversely, we identified 10 genes that were downregulated in Comparisons I and IV but upregulated in Comparisons III and VI, which may be involved in growth inhibition (Fig. 4d).

We overexpressed the 16 selected genes using the strong promoter *TEF1p* and individually knocked out 10 genes from the other comparison groups in ZQS18. Overexpression of *THI7* or knockout of *CSM4*, *PHO89*, *SDD1*, or *RGI2* significantly reduced the lag phase (Fig. 4e) without affecting the maximum optical density at 600 nm (OD$_{600}$) (Fig. 4f). Therefore, the relationship between these five genes and cell metabolism was further analysed. THI7, a plasma membrane transporter protein for thiamine uptake, is associated with EMP flux and fermentation efficiency[36], with its overexpression enhancing

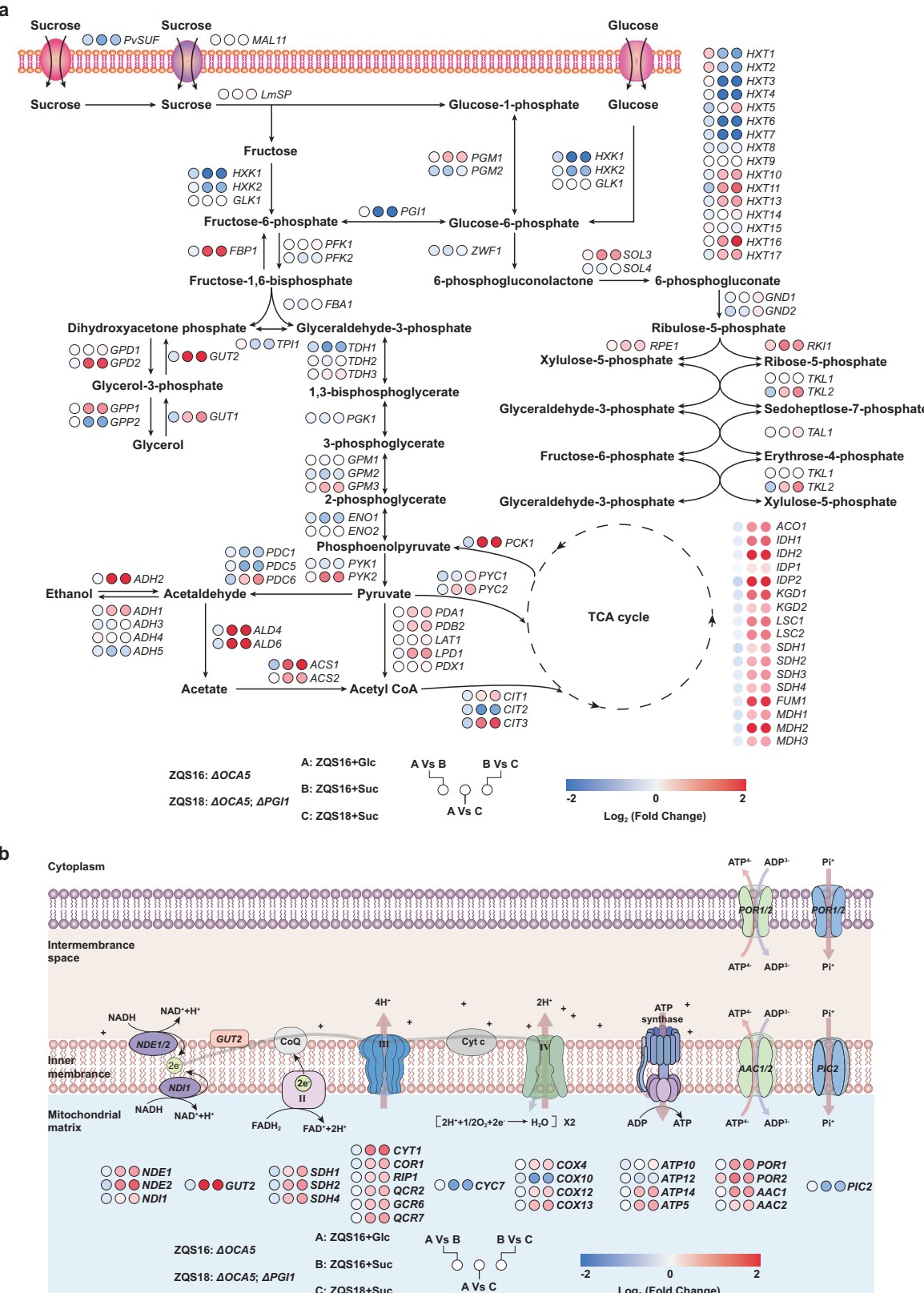

**Fig. 3 | Transcriptomic characteristics of *PGI1* knockout Crabtree-negative strains in central carbon metabolism and respiratory chain. a** Metabolic flux analysis of central carbon metabolism using the transcriptomes of ZQS16+Glc, ZQS16+Suc, and ZQS18+Suc. The log₂ fold change in gene expression (down/up) is represented by blue (downregulation) and red (upregulation) dots. The intensity of the red colour indicates the degree of gene upregulation, with darker shades indicating more pronounced upregulation. Conversely, darker shades of blue indicate a more pronounced downregulation of genes. **b** Flux and gene expression analyses of the electron transport chain and ATP synthase.

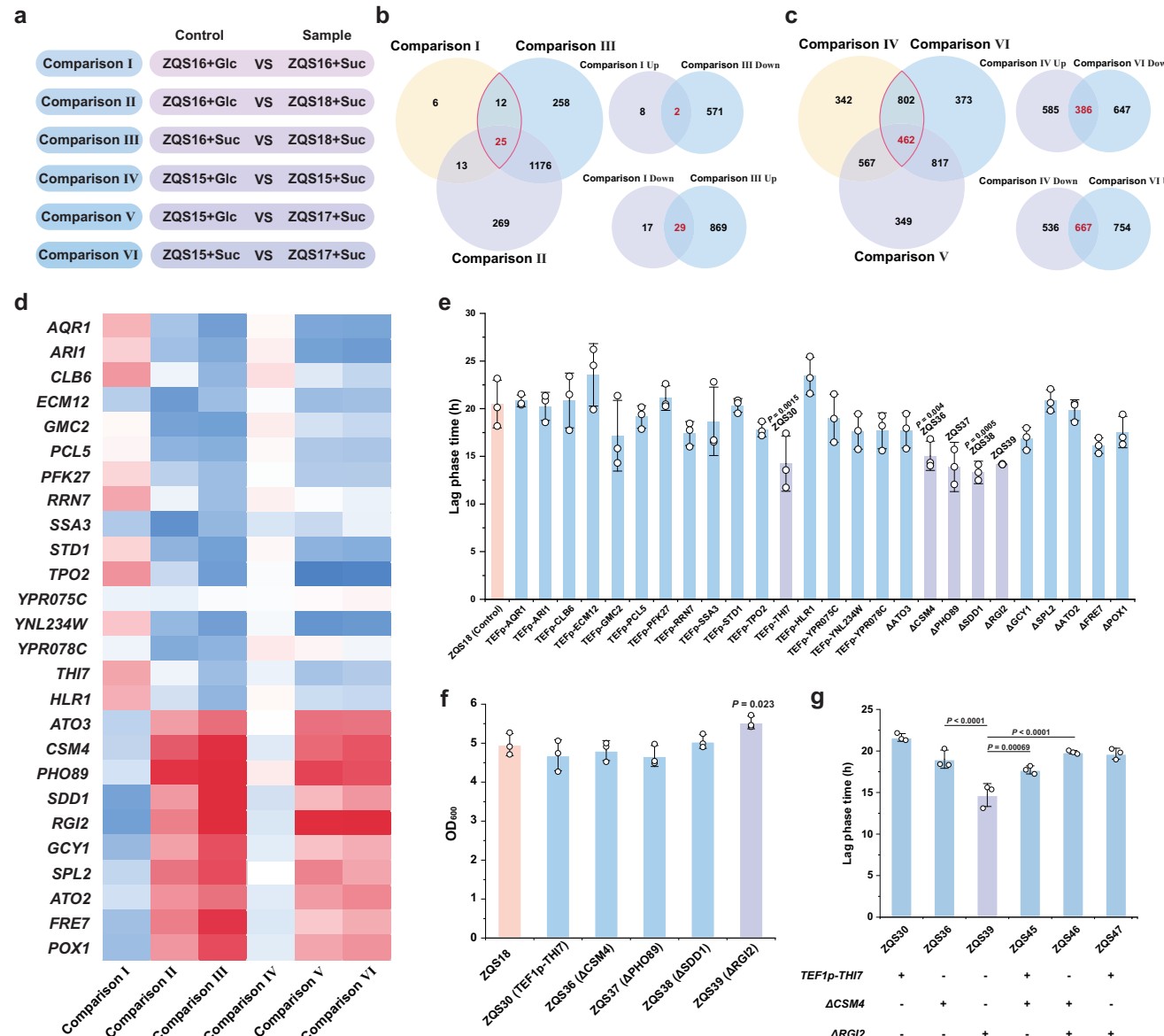

**Fig. 4 | Excavation and identification of growth-regulated genes. a** Grouping of different comparative strains. The genotypes of strains: ZQS16 (*OCA5Δ*), ZQS17 (*PGI1Δ*), ZQS18 (*OCA5Δ; PGI1Δ*). Comparison I and Comparison IV for sucrose phosphorolysis metabolism mode versus glucose metabolism mode. Comparison II and Comparison V were Crabtree-positive strains in the glucose metabolism pattern versus Crabtree-negative strains in the sucrose metabolism pattern. Comparison groups III and VI were Crabtree-positive versus Crabtree-negative strains in sucrose metabolism patterns. **b** Venn plots showing genes in the three comparison groups of the *OCA5* knocked-out strains (Comparisons I, II, and III). **c** Venn plots showing genes in three comparison groups of *OCA5* wild-type strains (Comparisons IV, V, and VI). **d** Heatmap illustrating the upregulation and downregulation of transcription levels for the selected 26 genes across different comparison groups.

The log$_2$ fold change of genes (down/up) is represented by blue (down) and red (up). The intensity of the red colour indicates the degree of upregulation, with darker shades indicating more pronounced upregulation. Conversely, darker shades of blue indicate a more pronounced downregulation of genes. **e** Impact of overexpression or knockout of 26 potential growth-regulating genes on the lag phase of the strains. The untreated strain ZQS18 was the control. **f** Maximum OD$_{600}$ for strains with effectively shortened lag phase time. ZQS18 was the control strain. **g** Comparison of lag phase duration across strains with different gene knockout or overexpression combinations. The strains with single-factor treatment were controls. All data are expressed as mean ± SD of biological triplicates. Statistical analysis was conducted using the Student's *t*-test (two-tailed; sample size, *n* = 3). Source data are provided as a Source Data file.

central carbon metabolism. CSM4, involved in meiosis and telomere-mediated prophase motility[37], affects mitosis and can retard haploid yeast growth[38], suggesting that knocking out *CSM4* might have the opposite effect. RGI2, implicated in ethanol accumulation, slows ethanol production and enhances respiration when knocked out[39]. PHO89, a Na$^+$/Pi cotransporter protein[40], and SDD1, involved in ribosome-associated quality control[41], have unclear specific roles in cell metabolism. Additionally, overexpression or knockout of these genes in strains C800 and ZQS15 produced consistent results, highlighting the universal role of THI7, CSM4, and RGI2 in growth

regulation (Supplementary Fig. 9). The effects of different combinations of *THI7*, *CSM4*, and *RGI2* on the lag phase in ZQS18 were then investigated by creating strains ZQS45–ZQS47 with these gene combinations. Compared to strains treated with single factors, the combination strains exhibited a longer lag phase than ZQS39 (*RGI2Δ*) (Fig. 4g). Consequently, we selected ZQS39, which showed a 30.8% reduction in the lag phase, as the starting strain for further experiments. The significant reduction in the lag phase observed in ZQS39 (*RGI2Δ*) suggests that *RGI2* plays a crucial role in regulating metabolic adaptation during the early growth phase. This effect may be

attributed to several factors associated with central carbon metabolism (Supplementary Fig. 10). A comparative analysis between ZQS39 (*RGI2Δ*) and ZQS18 revealed that the NADH/NAD$^+$ ratio in ZQS39 was lower (Supplementary Fig. 10a), suggesting an enhanced redox state. Notably, the NADPH/NADP$^+$ ratio remained unchanged (Supplementary Fig. 10b), which implies that the PPP was not significantly affected by *RGI2* deletion. In parallel, the ATP content in ZQS39 increased (Supplementary Fig. 10c), supporting the hypothesis that *RGI2* deletion accelerates energy production by favouring glycolysis over respiration. These observations imply that *RGI2* influences the metabolic balance between glycolysis and the TCA cycle, potentially reducing competition between these pathways. Overall, these findings indicate that *RGI2* deletion facilitates a more efficient metabolic transition, ultimately contributing to the observed reduction in lag phase duration.

## Cytoplasmic synthetic energy system regulates ATP production

Yeast oxidises NADH via five primary pathways[33]: (1) ethanol fermentation, (2) glycerol production, (3) cytoplasmic NADH respiration via mitochondrial outer membrane NADH dehydrogenases NDE1 and NDE2, (4) glycerol-3-phosphate shuttling for cytoplasmic NADH respiration, and (5) oxidation of mitochondrial NADH by the inner membrane NADH dehydrogenase NDI1 (Supplementary Fig. 11a). Among these, only the ethanol and glycerol pathways occur in the cytosol. In strain ZQS39, the lack of ethanol and glycerol accumulation disrupted NADH re-oxidation within the cytoplasm. To enhance the utilization of cytoplasmic NADH for ATP synthesis, we overexpressed *NDE1* and *NDE2* to compensate for the deficiency in NADH reoxidation[42]. However, this approach caused excessive oxidation of cytosolic NADH in the respiratory chain, ultimately leading to cell death[43] (Supplementary Fig. 12a, b). To address this issue, we introduced a transhydrogenase (AcTH) from *Azotobacter chroococcum* to facilitate NADPH-to-NADH conversion[17] (Fig. 5a). Strain ZQS48, engineered to overexpress *NDE1*, *NDE2*, and *AcTH* (the synthetic energy system), exhibited normal growth in sucrose medium (Fig. 5b and Supplementary Fig. 12c) and an enhanced capacity for ATP production (Fig. 5c).

This synthetic energy system enables ATP production through cytoplasmic NADH, supporting cell growth even when the TCA cycle is insufficient for energy production. To further investigate this, we downregulated the TCA cycle by replacing the *IDH2* (subunit of the mitochondrial NAD($^+$)-dependent isocitrate dehydrogenase) promoter with weaker alternatives[44] (Fig. 5d). In ZQS39, which lacks the synthetic energy system, growth progressively declined as promoter strength decreased, with ZQS53 eventually arresting growth (Fig. 5e). In contrast, ZQS48, which contains the synthetic energy system, alleviated growth restriction (Fig. 5f), demonstrating that the synthetic system can compensate for reduced TCA cycle function. However, ZQS48 displayed a partial resurgence in ethanol and glycerol production (Fig. 5g, h), suggesting that a fraction of NADH was oxidised to NAD$^+$ via overflow metabolism. We hypothesised that the presence of AcTH causes excessive NADH accumulation, forcing cells to rely on overflow metabolism to maintain the NADH/NAD$^+$ balance in the cytoplasm. To address this, we introduced phosphoketolase (LmPK) from *L. mesenteroides*[45] and phosphate acetyltransferase (CkPTA) from *Clostridium kluyveri*[46], enabling the PPP to synthesise acetyl-CoA via the PTA pathway, which feeds into the TCA cycle (Fig. 5a). This modification promoted aerobic respiration over ethanol production. The resulting strain ZQS59 effectively reduced ethanol accumulation (Fig. 5g). To further eliminate NADH consumption via ethanol fermentation and glycerol production, we progressively knocked out alcohol dehydrogenase genes (*ADH1*, *ADH3*) and glycerol-3-phosphate phosphatase genes (*GPP1*, *GPP2*) in ZQS59, producing strains ZQS60–ZQS65. Blocking ethanol synthesis in ZQS62 led to significant glycerol accumulation (Fig. 5h), while blocking glycerol production in

ZQS61 resulted in increased ethanol accumulation (Fig. 5g). This pattern indicates that cytoplasmic NADH preferentially supports substrate-level phosphorylation. In strain ZQS65, ethanol and glycerol levels were reduced to minimal (Fig. 5g, h) without compromising growth, with a $\mu_{max}$ of 0.20 h$^{-1}$ (Supplementary Fig. 11b, c). Compared to ZQS39, ZQS65 maintains a lower NADPH/NADP$^+$ ratio (Supplementary Fig. 11d) while keeping a constant NADH/NAD$^+$ ratio, indicating that the synthetic energy system efficiently converts NADPH into NADH to sustain cytosolic NADH/NAD$^+$ balance. Additionally, the higher ATP content in ZQS65 (Fig. 5c) suggests that the modified synthetic energy system enables yeast to bypass fermentation, accelerating NADH flux into the respiratory chain and relying solely on oxidative phosphorylation for energy production. To further validate the fermentation performance of strain ZQS65, we conducted batch fermentation in a 5-L fermenter for both ZQS65 and ZQS18. The results demonstrated that ZQS65, incorporating the synthetic energy system, achieved a $\mu_{max}$ of 0.24 h$^{-1}$, a maximum biomass yield of 0.24 g DCW/g sucrose, and a sucrose uptake rate of 0.17 g/g DCW/h (Supplementary Fig. 13a, c). In comparison, ZQS18 exhibited slightly lower values (0.22 h$^{-1}$, 0.24 g DCW/g sucrose, 0.16 g/g DCW/h), highlighting the superior growth performance of ZQS65. Notably, while ZQS18 ceased sucrose consumption after reaching maximum biomass at 48 h, ZQS65 continued to gradually utilize sucrose until it was completely depleted, even after reaching maximum biomass at the same time point (Supplementary Fig. 13a, b). Metabolite profiling revealed distinct byproduct accumulation patterns between the strains (Supplementary Fig. 13a, b). In ZQS65, ethanol and acetate concentrations began increasing significantly after 48 h (reaching 0.72 g/L and 0.53 g/L, respectively), while glycerol and pyruvate remained below detection limits throughout. This contrasts sharply with ZQS18, where all measured metabolites (ethanol, acetate, glycerol, pyruvate) maintained at negligible levels (<0.5 g/L) during the entire fermentation process (Supplementary Table. 1). These observations confirm that ZQS65 exhibits minor but detectable overflow metabolism during prolonged sucrose utilization. We attribute this metabolic shift to the engineered energy system's altered redox balance and ATP yield. Additionally, we observed that both strains ZQS18 and ZQS65 exhibited pyruvate accumulation below 0.01 g/L during fermentation, yet showed minimal overflow metabolism. This suggests that most of the pyruvate entered the TCA cycle to fuel respiratory metabolism.

## Synthesis of non-ethanol products in the Crabtree-negative strain with an optimized synthetic energy system

The primary objective of constructing engineered strains is to enhance the carbon atom economy in the biosynthesis of chemicals and materials. Crabtree-negative microorganisms offer a distinct advantage in natural product synthesis due to their efficient energy utilization and minimal byproduct formation, including organic acids and secondary metabolites[44,47]. Here, we demonstrated the advantages of the engineered strains, ZQS39 and ZQS65, for natural product synthesis by validating their performance. Four specific products (lactic acid, 3-HP, *p*-coumaric acid, and farnesene) were selected to validate this model. Lactate is derived from pyruvate, while 3-HP originates from acetyl-CoA (Fig. 6a). *p*-coumaric acid and farnesene serve as representative products of the shikimate and mevalonate (MVA) pathways, respectively (Fig. 6a). These four products were selected to evaluate the metabolic flux distribution in strain ZQS39 and ZQS65, specifically assessing its capacity for pyruvate and acetyl-CoA supply, as well as its efficiency in channelling carbon flux toward the shikimate and MVA pathways for product synthesis.

Lactic acid, a valuable three-carbon hydroxycarboxylic acid derived from pyruvate and dependent on NADH, has diverse applications in the food, cosmetics, and agriculture industries[48] (Fig. 6b). Given that yeast lacks lactate dehydrogenase, which is crucial for lactic acid synthesis, we introduced the *LlLDH* gene from *Lactococcus lactis*

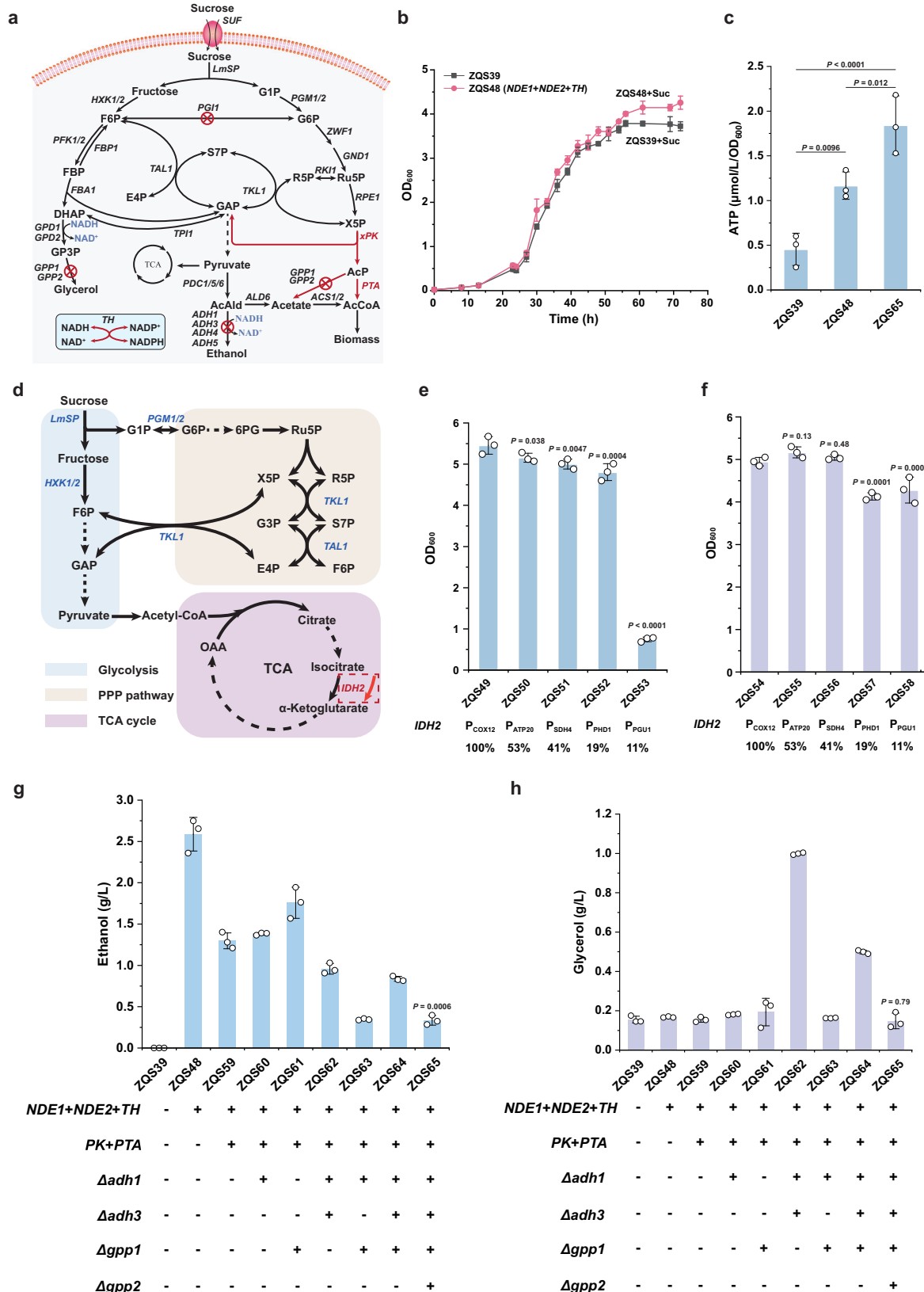

into strains ZQS15, ZQS39 and ZQS65, creating strains ZQS66, ZQS67 and ZQS68, respectively. ZQS68, a Crabtree-negative strain with a synthetic energy system, produced 3.29 g/L of lactic acid with a yield of 0.22 g/g sucrose, an 11-fold increase over the control strain ZQS66 (Fig. 6c), confirming its efficiency in lactic acid production. However, the lactate synthesis ability of ZQS68 is lower than that of ZQS67

(Fig. 6c), which lacks the synthetic energy system. This indicated that the introduction of the synthetic energy system consumes cytosolic NADH, thereby competing with lactate synthesis. In traditional Pdc⁻ based Crabtree-negative strains, the same strategy results in a lactic acid titre of 3.20 g/L, which is lower than the 4.99 g/L achieved by ZQS67[10]. Next, we explored 3-HP, a promising third-generation

**Fig. 5 | Enhanced energy metabolism through reoxidation of NADH in cytoplasmic. a** Schematic diagram showing reconfiguration of the central carbon metabolism network in yeast. Red lines indicate introduced exogenous pathways, while knocked-out genes are marked with crosses. **b** Growth of strain (ZQS48) containing the synthetic energy system, with ZQS39 as the control strain. **c** Comparison of ATP content in strains with the synthetic energy system. ZQS39 is the control strain without the synthetic energy system, ZQS48 (*NDE1 + NDE2 + TH*) contains the synthetic energy system, and ZQS65 (*NDE1 + NDE2 + TH; PK + PTA; ADH1Δ; ADH3Δ; GPP1Δ; GPP2Δ*) contains the synthetic energy system with optimised central carbon metabolism. **d** Schematic diagram illustrating the metabolic connections between glycolysis, the PPP, and the TCA cycle. Red arrows indicate

the downregulation of *IDH2* gene expression, leading to reduced TCA cycle flux. **e** Growth of ZQS39, lacking the synthetic energy system, after replacing *IDH2p* with weaker promoters. IDH2 was expressed under a series of promoters, including *COX12p* (100%), *ATP20p* (53%), *SDH4p* (41%), *PHD1p* (19%), and *PGU1p* (11%). **f** Growth of ZQS48, containing the synthetic energy system, after replacing *IDH2p* with weaker promoters. **g** Ethanol accumulation during energy metabolism remodelling. **h** Glycerol production during energy metabolism remodelling. All data are expressed as mean ± SD of biological triplicates. Statistical analysis was conducted using the Student's *t*-test (two-tailed; sample size, *n* = 3). Source data are provided as a Source Data file.

bio-based compound pivotal for synthesising propionic acid, bioplastics, and other materials[49]. The biosynthesis of 3-HP requires malonyl-CoA as a precursor and is catalysed by malonyl-CoA reductase (MCR)[50] (Fig. 6d). To initiate 3-HP production, we employed a split mutant of the bifunctional enzyme gene *CaMCR* (*CaMCR-C*[N941V K1107W S1115R] and *CaMCR-N*) from *Chloroflexus aurantiacus*[51]. The engineered strain ZQS71 produced 3-HP at a titre of 1.96 g/L with a yield of 0.15 g/g sucrose, which is 8.2-fold higher than that of ZQS69 and 3.1-fold improvement compared to ZQS70 (Fig. 6e). This result indicated that Crabtree-negative strains equipped with the synthetic energy system exhibit an enhanced supply of acetyl-CoA for the synthesis of their derivatives. In addition, when compared to a glucose-metabolizing strain under the same strategy, ZQS71 achieved 1.96 g/L of 3-HP, which is significantly higher than the 0.31 g/L produced by the glucose-metabolizing strain[49]. This further underscores the superior performance of our engineered strain in facilitating efficient 3-HP production.

To further evaluate flux levels in the shikimate and MVA pathways of the engineered strains, we targeted the production of *p*-coumaric acid and farnesene. The shikimate pathway produces *p*-coumaric acid downstream of L-phenylalanine or L-tyrosine (Fig. 6f)[52,53]. We introduced *FjTAL* from *Flavobacterium johnsoniae* to convert L-tyrosine into *p*-coumaric acid and assessed the titres in strains ZQS72, ZQS73 and ZQS74. ZQS74 produced 215.47 mg/L of *p*-coumaric acid, with a yield of 14.13 mg/g sucrose, doubling the titre of the control strain ZQS72 (Fig. 6g). The result indicated that the engineered strain retains a competitive edge in the shikimate pathway. However, we hypothesize that the suboptimal yield enhancement of *p*-coumaric acid, compared to 3-HP, could be attributed to its more extended biosynthetic route, which may introduce inefficiencies such as metabolic flux dilution. Nevertheless, the introduction of *FjTAL* in strain ZQS65 still resulted in a higher *p*-coumaric acid titre compared to the previously reported titre of 12.90 mg/L under the same strategy with glucose metabolism[52]. Additionally, yeast natively utilises the MVA pathway to produce isopentenyl diphosphate and dimethylallyl diphosphate, which are essential precursors of farnesene synthesis[54] (Fig. 6h). We introduced *MdAFS* from *Malus domestica* into ZQS15, ZQS39 and ZQS65, generating strains ZQS75, ZQS76 and ZQS77, respectively. Strain ZQS77 achieved a farnesene titre of 105.27 mg/L, representing a 22.0% increase in titre and a 77.0% yield improvement over the control strain ZQS75 (Fig. 6i). These findings underscore the advantages of Crabtree-negative strains with optimised synthetic energy systems for the synthesis of pyruvate or malonyl-CoA derivatives and compounds produced via the shikimate or MVA pathways when using sucrose as the carbon source. This establishes a robust foundation for the efficient production of organic acids, terpenes, flavonoids, and other valuable compounds.

## Discussion

Carbon is a crucial resource for the production of organic compounds; however, it is often used inefficiently and in ways that are both wasteful and harmful to the environment[55]. In industrial biotechnology, the efficient use of carbon in metabolic pathways directly influences the cost-effectiveness of substrates and, consequently, the final cost of a

product[56]. This study describes the development of a yeast chassis cell optimised for high carbon atom economy. Initially, sucrose, an inexpensive carbon source, was selected as the substrate. Replacement of the yeast's endogenous hydrolase with sucrose phosphorylase (LmSP) effectively eliminated glucose repression. Unlike hydrolysis, sucrose phosphorolysis generates fructose and glucose-1-phosphate, which seamlessly integrate into central carbon metabolism, enhancing energy production (Fig. 1a). Next, the Crabtree effect was eliminated by knocking out the *OCA5* and *PGI1* genes (Fig. 2c, g), allowing carbon atoms to fully enter the TCA cycle and respiratory chain for energy metabolism. Additionally, we implemented a synthetic energy system to regenerate NAD⁺ in the cytoplasm, addressing the potential cytoplasmic NADH/NAD⁺ imbalance caused by ethanol and glycerol pathway degradation (Fig. 5a). Ultimately, this engineered system demonstrated the ability to synthesise various compounds in significantly higher yields when sucrose was used as the carbon source, compared to the glucose-based metabolism model (Fig. 6). Our findings confirm that carbon-conserving metabolic pathway strategies can achieve high carbon yields in yeast cell factories.

During microbial biosynthesis, the change in Gibbs free energy as substrates are converted into products is mainly conserved as ATP or used to drive biochemical reactions[26]. The efficiency of energy conservation in central metabolism, particularly the conversion of ADP and phosphate to ATP, significantly influences product yield[28]. Therefore, we explored ways to enhance energy conservation during yeast growth on sucrose. In wild-type yeast, sucrose undergoes hydrolysis, leading to a loss of free energy released during this process. In contrast, sucrose phosphorolysis conserves this energy by forming phosphorylated intermediates. We introduced PvSUF, an engineered variant of the sucrose transporter protein that facilitates the transport of sucrose into cells without requiring ATP, unlike the natural MAL11 transporter (Fig. 1a). This approach effectively reduced free energy loss and increased ATP content (Fig. 1h). Furthermore, compared to glucose metabolism, sucrose phosphorolysis mitigated the Crabtree effect (Fig. 2a). We propose that the rate-limiting phosphorylation in sucrose phosphorolysis restricts the flux through EMP, diminishing the overflow metabolism of pyruvate.

The Crabtree effect is triggered by metabolic overflow at the pyruvate node (short-term effect) and glucose-driven respiratory enzyme inhibition (long-term effect)[13]. While this effect has evolutionary benefits for yeasts, industrial yeast strains used for chemical production do not benefit. Ethanol produced from carbon sources via the Crabtree effect reduces the flux of desired chemical products. Two critical strategies are essential to address this issue: developing a novel pathway to replace ethanol production with NADH oxidation and addressing the supply of cytosolic acetyl-CoA. We first targeted the inositol pyrophosphatase gene *OCA5*, which regulates the balance between the EMP pathway and TCA cycle[17] under sucrose phosphorolysis conditions (Fig. 2c). This approach partially alleviated the Crabtree effect (Fig. 2e, f). We hypothesised that the excessive utilization of carbon sources, leading to high EMP flux, significantly contributes to the Crabtree effect. To test this, the crosstalk between the PPP and EMP pathway was disrupted by knocking out the *PGI1* gene

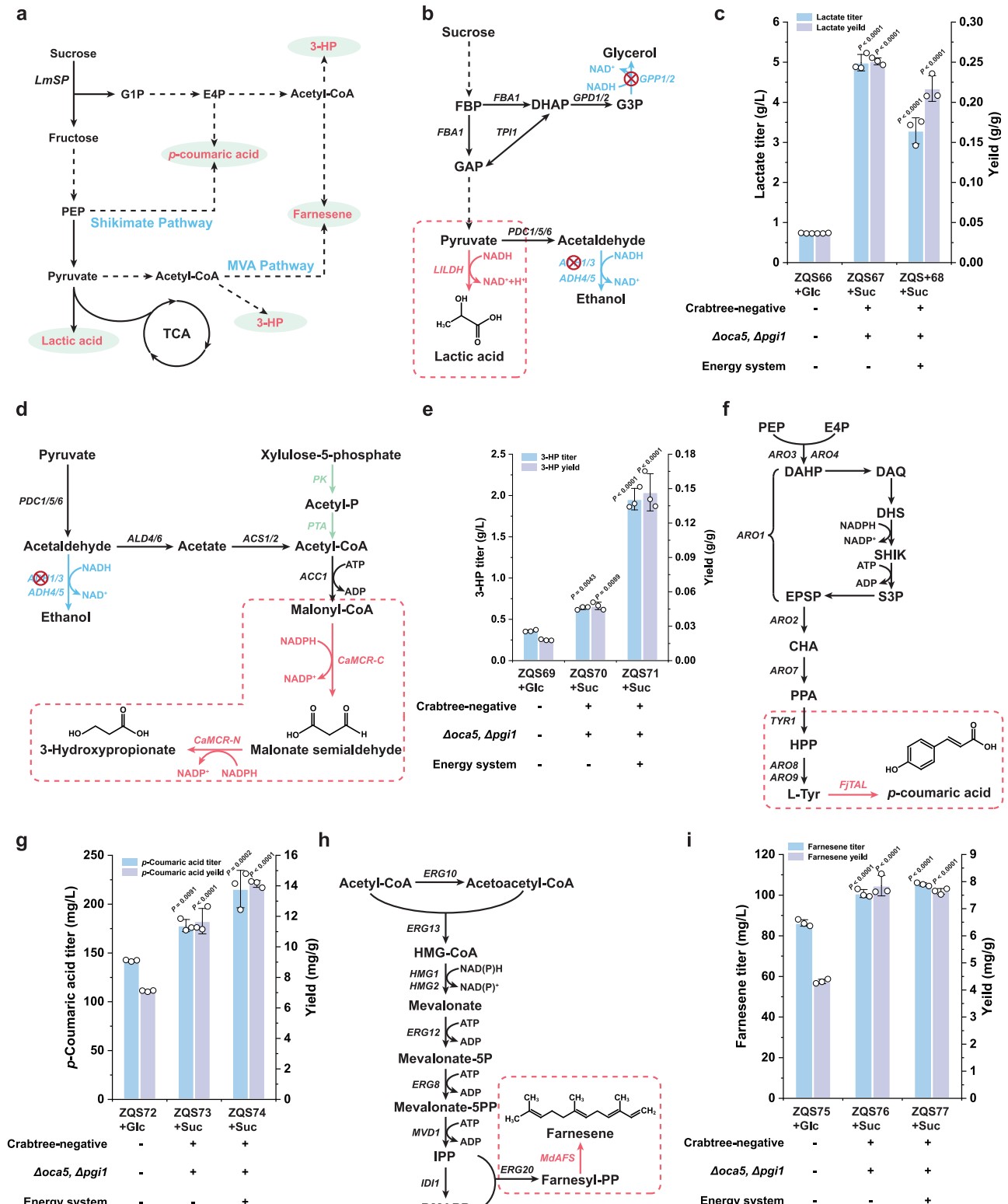

**Fig. 6 | The Crabtree-negative strain with the synthetic energy system was applied to the synthesis of compounds through different pathways. a** Pathway from sucrose phosphorolysis to different products, including lactic acid, 3-hydroxypropionic acid (3-HP), *p*-coumaric acid, and farnesene. The red font indicates various products, while the blue font denotes the pathways through which these products are synthesised. **b** Lactic acid synthesis pathway. Blue font indicates competing pathways. **c**. Titre and yield of lactic acid produced by the engineered strain ZQS67 and ZQS68, with ZQS66 as the Crabtree-positive control strain utilising sucrose phosphorolysis. **d** 3-HP synthesis pathway. Blue font indicates competing pathways, and green font indicates introduced exogenous pathways. **e** Titre and yield of 3-HP produced by the engineered strain ZQS70 and ZQS71, with ZQS69 as the Crabtree-positive control strain utilising sucrose phosphorylation. **f** *p*-coumaric acid synthesis pathway. **g** Titre and yield of *p*-coumaric acid produced by the engineered strain ZQS73 and ZQS74, with ZQS72 as the Crabtree-positive control strain utilising sucrose phosphorylation. **h** Farnesene synthesis pathway. **i** Titre and yield of farnesene produced by the engineered strain ZQS76 and ZQS77, with ZQS75 as the Crabtree-positive control strain utilising sucrose phosphorylation. All data are expressed as mean ± SD of biological triplicates. Statistical analysis was conducted using the Student's *t*-test (two-tailed; sample size, *n* = 3). Source data are provided as a Source Data file.

(Fig. 2g) to reduce EMP flux. The *PGI1Δ* strain no longer produced ethanol, indicating a successful elimination of the Crabtree effect (Fig. 2h). Our analysis revealed that reduced EMP flux (confirmed using transcriptomic analysis) was the primary factor behind the significantly lower NADH production in the cytosol compared to the glucose metabolism mode. In this mode, residual NADH could be oxidised via the respiratory chain by mitochondrial external NADH dehydrogenases (NDE1 and NDE2). Since the downstream metabolism of pyruvate was not disrupted, it was redirected toward cytosolic acetyl-CoA synthesis and respiration, keeping the acetyl-CoA synthesis pathway active. Thus, in the sucrose phosphorolysis metabolic mode, the *PGI1Δ* strain meets the requirements for NADH oxidation and cytosolic acetyl-CoA supply, resulting in yeast transition to a Crabtree-negative phenotype.

Redox metabolism is a finely tuned process characterised by precise stoichiometric ratios and coordinated enzyme activities[57]. Different substrates can impose varying constraints on this balance, potentially leading to an imbalance and metabolic intermediates or ATP wastage[6]. To address these challenges, we engineered an energy synthesis pathway to boost ATP production (Fig. 5a). This synthetic energy system regenerated cytosolic $NAD^+$ through the respiratory chain, enhancing ATP production (Fig. 5c) and partially replacing the TCA cycle (Fig. 5f). However, this led to a partial resumption of ethanol and glycerol accumulation (Fig. 5g). We hypothesise that introducing transhydrogenase (TH) into the synthetic energy system disrupts the $NADH/NAD^+$ balance in the cytosol, prompting the cells to reactivate ethanol and glycerol production as a compensatory response. Two strategies can be used to eliminate ethanol accumulation: redistributing the cytoplasmic energy supply to alleviate metabolic pressure or completely truncating the ethanol synthesis pathway. We addressed the former by introducing the PTA pathway, which reduced ethanol accumulation (Fig. 5a). Additionally, we knocked out the genes *ADH1*, *ADH3*, *GPP1*, and *GPP3*, which led to a significant reduction in ethanol and glycerol production without affecting cell growth. This finding further demonstrates that even without ethanol overflow metabolism, the synthetic energy system maintains the cytoplasmic $NADH/NAD^+$ balance, replacing the ethanol synthesis pathway.

Our findings demonstrate that the Crabtree-negative strain incorporating the synthetic energy system has a significant advantage in producing target chemicals. This approach optimises carbon utilization and enhances energy synthesis, establishing a highly efficient chassis to produce platform chemicals. Reprogramming energy metabolism highlighted the intricate metabolic regulation in yeast, pinpointing crucial balance points within this regulatory network. The flexible organisation of cell structures and the reconfiguration of central carbon metabolism illuminated these balance points and deepened our understanding of the fundamental principles governing cell metabolism.

## Methods
### Strains, reagents, and media
As a derivative strain of *S. cerevisiae* CEN.PK2-1D, strain C800 (*MATα*; *ura3-52*; *trp1-289*; *leu2-3,112*; *his3Δ1*; *MAL2-8C*; *SUC2*; *gal80Δ*)[58] was used as the experimental model for this study. All derived strains used in this study and the corresponding genotypes were listed in Supplementary Data 2. *Escherichia coli* JM109 was used for the construction and amplification of CRISPR/Cas9 plasmids or donor DNA. The CloneMulti One-Step Cloning Kit used for plasmid construction was purchased from Vazyme Bio (Nanjing, China), and the Golden Gate Assembly Kit used for multi-gene editing plasmid construction was purchased from New England Biolabs (MA, USA). The enzyme Phanta Max Master Mix (Dye Plus) for PCR amplification was purchased from Vazyme Bio (Nanjing, China). Plasmid mini-prep and DNA gel purification kits were purchased from Vazyme Bio (Nanjing, China). All chemical standards were purchased from Yuanye Bio (Shanghai, China) unless otherwise stated. The 20×YNB (Yeast Nitrogen Base) medium was purchased from Sangon Bio (Shanghai, China). Yeast extract and peptone were purchased from Thermo Fisher Scientific (Shanghai, China), and other reagents for strain culture were purchased from Sinopharm (Beijing, China).

Luria-Bertani (LB) medium (10 g/L peptone, 5 g/L yeast extract, and 10 g/L NaCl) supplemented with 100 μg/mL ampicillin was used for the cultivation of *E. coli*. YPD medium (20 g/L peptone, 10 g/L yeast extract, and 20 g/L glucose) was used for the cultivation and fermentation of *S. cerevisiae*; for growth on sucrose, glucose was replaced with 40 g/L sucrose. YNB medium (1.74 g/L yeast nitrogen base without amino acids, 5 g/L ammonium sulfate, and 20 g/L glucose or 40 g/L sucrose) was used for selecting transformed yeast. Supplements of 50 mg/L leucine, 50 mg/L histidine, 50 mg/L tryptophan, and 20 mg/L uracil were added as required.

### Genetic manipulations
All primers and codon-optimized genes used in this study were synthesized by Sangon Bio (Shanghai, China). Native promoters, other genes, and terminators were cloned from the genomic DNA of *S. cerevisiae* C800. Detailed information about primers and genes is listed in the Supplementary Data 3 and 4. The plasmids used in this study are listed in Supplementary Data 5. The expression cassettes for genes deletion and integration were assembled by fusion PCR and performed using CRISPR/Cas9[59]. CRISPR/Cas9 plasmids were constructed using ClonExpress Multis One-Step Cloning Kit. The Golden Gate Assembly Kit was used to construct multi-gene editing CRISPR/Cas9 plasmids[60]. All fragments and plasmids were confirmed by Sanger sequencing (Sangon Biotech). For gene deletion and integration, all sgRNAs (20 bp gRNA targeting sequence) were selected according to the CHOPCHOP website (http://chopchop.cbu.uib.no/) and their sequences are listed in Supplementary Data 6. Plasmids and fragments were transfected in *S. cerevisiae* with the lithium acetate transformation method[61]. After transformation, cells were plated on selective medium (containing amino acids) and incubated at 30 °C for 3–5 days.

### Strain cultivation and growth analyses
*E. coli* JM109 was used for plasmid extraction after 10 h of incubation in LB medium. *S. cerevisiae* was cultured with YPD medium in shake flasks for growth tests. A single colony was inoculated into a shake tube containing 3 mL YPD medium and grown for 16 to 30 h, depending on the growth rate of the modified strain, until reaching the logarithmic phase. The culture was then transferred to a 250-mL shake flask containing 25 mL YPD medium, with a final $OD_{600}$ of 0.02. The flask was incubated at 30 °C with shaking at 200 rpm. For strains containing plasmids, the initial cultivation was performed in YNB medium lacking the necessary amino acids. After 24 h, the culture was transferred to a 250-mL shake flask for subsequent growth tests. To achieve the loss of the Cas9 plasmid after knockout or integration, the correct transformants were picked and inoculated into shaker tubes containing 3 mL YPD medium. The cultures were then incubated until reaching logarithmic growth. Subsequently, the culture was streaked onto YPD plates containing 1 g/L 5-fluoroorotic acid to confirm plasmid loss. For fermentation of strains producing synthetic compounds, the seed culture was first transferred to 250 mL shake flasks containing 25 mL medium, achieving a final $OD_{600}$ of 0.02. Fermentation was carried out for 72-120 h, with periodic sampling to assess the titre of the target product. During the farnesene fermentation, 10% dodecane was added to the fermentation broth at 24 hours into the fermentation process for product extraction[62]. All strains mentioned above were cultured using their respective carbon sources, as indicated in the "Results" section. For example, strain C800 was cultured in YPD medium containing glucose, noted as "C800+Glc", and in medium containing sucrose, noted as "C800+Suc".

## Spotting assay

Strains were pre-cultured in shake flasks with the appropriate liquid medium to logarithmic growth phase. After rinsing the strain twice with distilled water, dilute it to $OD_{600} = 1.0$. Subsequently, perform a 10-fold serial dilution of the culture solution using distilled water. Pipette 10 µL of the diluted culture evenly onto YNB plates supplemented with four amino acids (leucine, histidine, tryptophan, and uracil) and incubate the plates at 30 °C for 3 days.

## Lag phase measurement

To measure the lag phase time, the growth curve of the strain needs to be plotted. The $OD_{600}$ values of the growth cycle were fitted using Origin 2018 64 bit. The lag phase time analysis was conducted in Rstudio using the GrowthRates package according to the instructions. Detailed calculation procedures and the code provided by T. Petzoldt can be found at Github (https://tpetzoldt.github.io/growthrates/doc/Introduction.html)[63].

## Maximum specific growth rate calculation

The maximum specific growth rate ($\mu_{max}$) was determined by analysing the growth curve during the exponential phase. A minimum of six data points were collected during the exponential growth phase to ensure accurate representation of the growth dynamics. The growth curve was fitted to a suitable model, specifically the Logistic model, which was chosen for its ability to accurately describe sigmoidal growth behaviour typical of microbial growth under controlled conditions. The differentiation was performed numerically using a central difference method to ensure accuracy, and the specific growth rate at each time point was derived from the slope of the fitted curve. The maximum slope, corresponding to the highest growth rate, was taken as the $\mu_{max}$. All measurements were performed in triplicate, and the results are reported as the mean ± standard deviation. Notably, this method may present $\mu_{max}$ as a single peak value rather than a sustained time interval due to: (1) noise amplification during differentiation of optical density data, particularly at metabolic transition points; (2) the algorithm's inherent sensitivity to rapid physiological transitions between growth phases; and (3) potential overestimation of maximum rates when analysis windows span multiple growth phases. While this approach provides high temporal resolution of growth dynamics, it differs from classical exponential phase analysis by capturing transient metabolic activation states rather than sustained exponential growth. Comparative validation with traditional curve-fitting methods confirmed that the sliding-window approach yields biologically plausible $\mu_{max}$ estimates.

## Transcriptome profiling

The test strains were cultured in 25 mL YPD medium (glucose or sucrose chosen as the carbon source depending on the strain) until $OD_{600} = 1.0$. This process was conducted in triplicate for biological replication. Following cultivation, cells were centrifuged at 6010 × g for 2 min at 4 °C to collect the biomass. The cell pellet was promptly quenched in liquid nitrogen and stored at −80 °C until RNA extraction. Total RNA extraction was performed by BioMarker company (Beijing, China), which also conducted and provided the original readings and differential expression analysis for each sample. Differential gene expression was analysed using edgeR[64], with Fold Change≥1.5 and P value < 0.05 as the selection criteria. GO and KEGG term enrichment analysis were performed using the ClusterProfiler R package[65].

## Analytical methods

Cell growth was determined by measuring the $OD_{600}$ of the cultures using a microplate reader Synergy H1 (BioTek, USA). Ethanol, glycerol, acetate, and pyruvate were quantified using a Shimadzu CBM-20A series HPLC equipped with a differential refractive index detector (RID-10A), with pyruvate additionally detected by a UV detector (210 nm).

The analysis utilized an HPX-87H column (Bio-Rad, USA) at 45 °C, with a mobile phase consisting of 5 mM sulfuric acid flowing at 0.6 mL/min. Prior to analysis, all fermentation samples were centrifuged at 13,523 × g for 2 min, and the clarified supernatant was filtered through a 0.22 µm membrane for testing.

For the detection of lactic acid, samples were taken from 24 to 96 h of fermentation. After centrifuging 1 mL sample at 13,523 × g for 2 min, the supernatant was filtered through a 0.22 µm filter membrane and stored at 4 °C for further use. Lactic acid was analysed using an HPX-87H column and a UV detector SPD-20A (210 nm) in the HPLC system. The column temperature was 50 °C. The mobile phase consisted of 5 mM sulfuric acid flowing at a rate of 0.6 mL/min[66]. For 3-HP detection, the pretreatment operation was consistent with lactic acid. The mobile phase was 0.5 mM sulfuric acid at a flow rate of 0.4 mL/min, the column temperature was 65 °C, and the detector was a UV detector (210 nm)[67].

For the detection of p-coumaric acid, 500 µL of the fermentation sample was mixed with an equal volume of methanol and vortexed for 5 min to extract p-coumaric acid. After centrifuging at 13,523 × g for 2 min, the supernatant was filtered through a 0.22 µm organic membrane filter and stored for further use[68]. C18 250 mm × 4.6 mm column (particle size 5 µm; Shimadzu, Japan) was used at a column temperature of 30 °C, with detection at a wavelength of 308 nm. The flow rate of the mobile phase was maintained at 1 mL/min. A gradient method was employed using two solvents: 0.1% formic acid in water (A) and acetonitrile (B). The programme started with 10% solvent B, transitioning from 10% to 40% (0–10 min), 40% to 50% (10–20 min), then returning to 10% (20–25 min), and finally holding at 10% for 5 min (25–30 min)[60].

Due to the addition of 10% dodecane during the fermentation of farnesene, 1 mL of the sample was taken for centrifugation during the pretreatment process. After centrifugation, 50 µL of dodecane phase was extracted and diluted 20-fold with methanol, followed by filtration through a 0.22 µm filter membrane. To quantify farnesene, a C18 column was employed with a mobile phase consisting of methanol and acetonitrile in a ratio of 7:3 (v/v), flowing at 0.8 mL/min. The column temperature was set at 40 °C, and detection was performed at 232 nm[69].

## Determination of ATP content

The ATP test kit was purchased from Biyotime (Shanghai, China). The test strains were cultured until $OD_{600} = 1.0$. Then, 1 mL of the culture was centrifuged at 6010 × g for 2 min at 4 °C. The supernatant was removed, and the cell pellet was washed twice with distilled water. Subsequently, 0.5 mm glass beads and 1 mL of lysis buffer were added. The mixture was vortexed for 5 min to fully lyse the cells. The sample was centrifuged at 13,523 × g for 5 minutes at 4 °C. In a 96-well shallow plate, 100 µL of ATP detection working solution was added first and incubated at room temperature for 3–5 min to consume all background ATP. Then, 20 µL of the lysate was added to the detection solution, and the Relative Light Units (RLU) were measured using a luminometer (all samples were tested in triplicate). ATP standard solutions of 0.01, 0.03, 0.1, 0.3, 1, 3, and 10 µM were prepared for the standard curve.

## Determination of the NADH/NAD$^+$ ratio

The NADH/NAD$^+$ test kit was purchased from Biyotime (Shanghai, China). Cells were cultured until $OD_{600} = 1.0$. The cells were then centrifuged and washed twice with distilled water. Next, 200 µL of lysis buffer and 0.5 mm glass beads were added, and the mixture was vortexed for 5 min to lyse the cells. The sample was then centrifuged at 13,523 × g for 5 min at 4 °C, and the supernatant was collected as the test sample. To determine the total amount of NAD$^+$ and NADH in the samples, 20 µL of the test sample was transferred to a 96-well shallow plate, followed by the addition of 90 µL of alcohol dehydrogenase

working solution. The plate was incubated at 37 °C in the dark for 10 min. Then, 10 μL of colour reagent was added to each well, mixed thoroughly, and incubated again at 37 °C in the dark for 10 min. Absorbance was measured at 450 nm. For the detection of NADH, the sample was heated in a water bath at 60 °C for 30 min to decompose NAD⁺. Then, 20 μL of the sample was transferred, and the same procedure was followed (all samples were tested in triplicate). NADH standard solutions were prepared at concentrations of 0.25, 0.5, 1, 2, 4, and 8 μM for the standard curve. The concentration of $NAD^+$ was calculated by subtracting the NADH content from the total amount of $NAD^+$ and NADH.

## Determination of the NADPH/NADP⁺ ratio

The NADPH/NADP⁺ test kit was purchased from Biyotime (Shanghai, China). Cells were cultured until $OD_{600} = 1.0$. The pretreatment of cells and the method of cell wall disruption were consistent with those used for NADH detection. After cell disruption, to measure the total amount of NADPH and NADP⁺ in the sample, 50 μL of the test sample was transferred to a 96-well plate. Then, 100 μL of glucose-6-phosphate dehydrogenase (G6PDH) working solution was added, followed by incubation at 37 °C in the dark for 10 min. Subsequently, 10 μL of chromogenic agent was added to each well, mixed thoroughly, and incubated again at 37 °C in the dark for 30 min. The absorbance was then measured at 450 nm. For the detection of NADPH, the sample was heated in a 60 °C water bath for 30 min to decompose NADP⁺. Then, 50 μL of the sample was taken and processed following the same steps (all samples were tested in triplicate).

## Bioreactor cultivation

For the fermentation in a 5-L bioreactor, a single colony of the engineered strain was inoculated into a 50 mL flask containing 5 mL of YPD medium and cultured for 24 hours at 220 rpm and 30 °C. Then, a 1% inoculum was transferred to a 250 mL seed flask containing 25 mL of YPD medium and cultured for another 24 hours at 220 rpm and 30 °C. Next, a 1% seed inoculum was transferred to a 5-L bioreactor containing 2.5 L of YPD medium, and cultured at 30 °C. The pH was maintained at 5.5 using 4 M NaOH. Aeration was set to 1 vvm, and the culture was agitated at 200–500 rpm to maintain dissolved oxygen above 20%.

## Reporting summary

Further information on research design is available in the Nature Portfolio Reporting Summary linked to this article.

## Data availability

Raw data of RNA-seq is deposited in the NCBI BioProject database under accession PRJNA1182066. Strain, primers, genes, plasmids, and gRNAs used in this work are provided in Supplementary Data 2–6. Source data are provided with this paper.

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

## Acknowledgements

This research was funded by the National Key Research and Development Programme of China (2022YFD2100804), the National Natural Science Foundation of China (32330084), the Key Research Project of Dongting Laboratory (2024-DTZD-001), Agricultural Science and Technology Innovation Fund of Hunan Province (2023CX29), Agricultural Science and Technology Innovation Fund of Hunan Province (2024CX02), and Changsha Municipal Natural Science Foundation (kq2502098). We appreciate Dr. Xiang Li from Chalmers University of

Technology for his assistance with data analysis; thank Guangjian Li from South China University of Technology for advice on manuscript revision.

## Author contributions

Z.X., Y.Z., J.L., and Y.S. designed this study; Z.X. performed most of the experiments, analysed the data and drafted the manuscript. Y.Z. assisted with parameter testing of Crabtree-negative strains. Y.W. and X.T. were involved in gene knockout and fermentation processes in yeast. S.Z., Q.L., F.H. and S.Z. assisted with yeast fermentation and data analysis. Z.X., L.W., J.M., J.L. and Y.S. wrote the manuscript; and all authors helped revise and approved the final version.

## Competing interests

The authors declare no competing interests.
