## [Peer Review file · Nature Communications]

Sucrose-driven carbon redox rebalancing eliminates the Crabtree effect and boosts energy metabolism in yeast.

Corresponding Author: Ms Juan Liu

Version 0:

Reviewer comments:

Reviewer #1

(Remarks to the Author)

The manuscript introduced the sucrose phosphorolysis pathway and deleted the phosphoglucose isomerase gene PGI1, and decoupling glycolysis from respiration and facilitating the metabolic transition of yeast to a Crabtree negative state. The results are interesting and will be useful for other metabolic engineering research. However, the quality of the manuscript needs to be improved. For example, the function of many enzymes was not described clearly. The result should be analyzed further.

Specific comments:

1. Line 109, what reaction do Mal12 and mal32 catalyze?
2. Line 117, what reaction does LmSP catalyze? Please provide the essential information.
3. What is PvSUF? Why the mutant version PvSUF I209F C265F G326C was used? Please at least give a brief introduction of the function of these proteins.
4. In figure 2A, what is ethanol yield rate? Ethanol yield and ethanol production rate are different.
5. It is necessary to measure the NADPH/NADP⁺ level, because pentose phosphate pathway should also be affected.
6. The products mentioned in the manuscript such as lactate, 3-HP, p-coumaric acid and farnesene, although the production of these chemical is improved, I think the reason of the improvement are different. Please discuss them in detail. For example, lactate and ethanol share the same precursor pyruvate, however, 3-HP and farnesene are derived from acetyl-CoA. Differently, p-coumaric acid requires E4P and PEP.
7. ZQS39 does not accumulated any ethanol, and the authors further engineered the strain to generate more NADH and more ATP. Are the engineered strains better than ZQS39? Besides from ATP generation, which parameters are better?
8. Please explain why in this engineered strain, the carbon can fully enter the TCA cycle and respiratory chain for energy metabolism.

Reviewer #2

(Remarks to the Author)

In this manuscript, the authors engineered the sucrose phosphorolysis and EMP pathway to restrict the importing sugar into the cell, coupling the reported inositol pyrophosphatase, a Crabtree-negative yeast was created alongside an obvious lag phase. Then reverse engineering was employed based on the transcriptional analysis, which shortened the lag phase to some extent. Furtherly, the redox equivalent balance between cytosolic and mitochondria was engineered, leading to ethanol and glycerol overflow and weaken cell growth. Efficient producing the non-ethanol chemicals demonstrated the advantage of this newly constructed yeast. This paper employed to a new strategy to shift *S. cerevisiae* from Crabtree positive to negative. Compared with the published paper, the novelty is not strong enough. The main reason is that the logic behind this strategy is the same as previous one: restricting sugar import and glycolysis to adjust the balance of redox equivalent between fermentation and respiration. In the other word, new molecular mechanism and knowledge was lacking, which is not supportive for publishing this work in Nat Comm at this stage.

Some suggestions may help to improve the quality of this work.

Major:

1. The specific growth rates of the engineered strains, especially the key one, are needed to be presented in this work. This parameter is quite important for describing the Crabtree effect related studies and comparison with reported works.
2. Since this new yeast showed superiority on producing non-ethanol chemicals, batch fermentations of the key strains such

as ZQS18 and ZQS65, etc. in bioreactor to characterize the specific growth rate, biomass yield, sugar uptake rate are suggested. This would be helpful for the community to get to know this strain for further study.

Minor:

1. How about the growth profiles of strains from 59-65?
2. About the lag phase, some of genes showed positive role, what is the explanation? More discussion is needed.
3. About the title, I don't think the carbon redistribution was driven by sucrose. And carbon redistribution may not be the main reason for eliminating Crabtree effect but the redox equivalent may play the essential role.

Reviewer #3

(Remarks to the Author)

The manuscript describes a massive amount of work, mainly involving recombinant DNA and the overexpression/deletion of several genes in the yeast *Saccharomyces cerevisiae*. The metabolic engineering strategies used had in part already been published by other groups before, but some seem to be new. At the end of the day, the authors claim to have generated a platform strain to produce different chemicals, which metabolizes sucrose mainly via respiration, leading to higher ATP yields and carbon availability, when compared to strains that preferably metabolize sugars via ethanolic fermentation. This, per se, can be considered a very interesting result.

There are some technical and conceptual details that could be addressed here, but I would like to mention three aspects that need clarification, before we can be sure that the work deserves publication as it is:

1) all comparisons made between the different strains, in terms of growth, are mainly based on the presentation of OD versus time graphs. I could not find calculations of specific growth rates, except on suppl. fig. 7. There is no indication on how these specific growth rates were calculated and I doubt whether one can perform good calculations, considering the small number of experimental points during the exponential growth phase in the growth curves presented throughout the work. I believe that a rigorous calculation of specific growth rates is important to compare the growth of different strains;

2) there is no mentioning of the AGT1 gene in the work. This is an important alpha-glucoside transporter that can transport sucrose. This gene is different from MAL11, although no distinction is made by the yeastgenome.org database. According to some colleague researchers, this is an error, which has been reported to SGD, but was never corrected. Here are two articles in which this distinction appears: Cousseau, F. E., Alves, S. L., Jr, Trichez, D., & Stambuk, B. U. (2013). Characterization of maltotriose transporters from the *Saccharomyces eubayanus* subgenome of the hybrid *Saccharomyces pastorianus* lager brewing yeast strain Weihenstephan 34/70. *Letters in applied microbiology*, 56(1), 21–29. <https://doi.org/10.1111/lam.12011>. AND

Trichez, D., Knychala, M. M., Figueiredo, C. M., Alves, S. L., Jr, da Silva, M. A., Miletti, L. C., de Araujo, P. S., & Stambuk, B. U. (2019). Key amino acid residues of the AGT1 permease required for maltotriose consumption and fermentation by *Saccharomyces cerevisiae*. *Journal of applied microbiology*, 126(2), 580–594. <https://doi.org/10.1111/jam.14161>

According to this expert with whom I discussed the issue, "AGT1 and MAL11 are different genes! The confusion comes from the "yeastgenomedatabase" which confused everything. As AGT1 is on chromosome VII, where the MAL1 locus is located, they mistakenly call it "MAL11". To <name hidden> (who cloned AGT1 for the first time) has already complained about it, I have also complained about it, but without success..... The MAL11 gene/transporter has been extensively studied by Rosario Lagunas, she even cloned and deposited the gene sequence in GenBank..... "

"in GenBank there are several entries for AGT1, one of them is CAA97322. The one for MAL11 is CAA10168. You will see that the sequences are different (I think 57% identity), and while MAL11 aligns with MAL31, MAL61 and MAL21 (the other transporters from the classic MAL loci), AGT1 forms a separate group"

It is important to understand whether the authors have paid attention to this or not, since the deletion of both MAL11 and AGT1 is presumably necessary to eliminate sucrose transport in *S. cerevisiae*.

3) Some important parameters displayed in the Figures are not clear, e.g. the "Ethanol yield rate" in Figure 2a. I understand what an Ethanol yield or a specific ethanol production rate are, but it is not clear to me what an Ethanol yield rate is.

Reviewer #4

(Remarks to the Author)

The manuscript describes the development of a Crabtree-negative yeast with enhanced energy system and its applications. The authors used sucrose as the carbon source and introduced sucrose phosphorylase to eliminate glucose repression and improve energy conservation. The authors conducted a comprehensive study to alleviate the Crabtree effect and boost energy production.

The following comments may help to improve the paper:

1. Many of the methods have been previously reported, e.g. sucrose using method in *Saccharomyces cerevisiae* (10.1016/j.ymben.2017.11.012), the effect of Oca5 deletion (10.1016/j.cell.2023.01.014), PGI1 deletion (10.1016/j.jbiotec.2013.11.025), and overexpressing of NDE1/2 has been a consensus to boost NADH oxidization. It would be better that the authors demonstrated more clearly the novelty of this work.
2. Other efforts to engineer a Crabtree-negative strain should be summarized; a table would be better.
3. Lines 131-134: The genotype of ZQS05 and the acquisition process of the mutant PvSUF are not very clear. It would be

more clear to also test the effect of mutant PvSUF alone.

4. Lines 215-218: More discussions on the rationale of the author's focus would help readers follow the article more easily.

5. It's impressive that Rgi2 helped boosting the growth and shorten lag phase. Further investigation of the critical role of Rgi2 in central carbon metabolism may enhance the novelty of this paper.

6. To further highlight the significance of their Crabtree-negative strain and optimized energy system, the authors should compare the production levels of their strains with those of other reported strains.

Version 1:

Reviewer comments:

Reviewer #1

(Remarks to the Author)

The authors have addressed my concerns, it can be accepted in its current form.

Reviewer #2

(Remarks to the Author)

The authors partly resolved my comments. To improve the quality of the work, the following concerns should be addressed.

1. Could the author explain the meaning of "carbon redox" in the title?

2. Line 68, the place to citing Fig S1 is confusing. It should be reported works but not a summary of this work.

3. The advantages and limitations are missing in Table S1. A proper summary of previous works should be added in the introduction, a table like this can not give the reader a clear overview of previous works and show the novelty of this work.

4. The specific growth rate (0.13 h⁻¹) of the strain C800 in YPD(glucose) is not convincing. Normally, under this condition, this value should be above than 0.3 h⁻¹. If this value is correct, no ethanol should be produced due to this value is much lower than the critical growth rate (0.26 h⁻¹ or 0.28 h⁻¹)(<https://doi.org/10.1073/pnas.0607469104>, Nat Commun 13, 2819 (2022)) of *Saccharomyces cerevisiae*.

5. Diauxic growth is a typical phenomenon of *Saccharomyces cerevisiae*, however this phenomenon was not observed in this study, specially under the sucrose condition. For example, fig 2h and 2k, we can find the obvious ethanol utilization after 45 h, but no corresponding cell growth presented. Please explain this.

6. Line 193-197, if the strains possess similar growth rate, why ZQS17 and 18 have extended cell growth cycle?

7. Line 227, what method was taken to do the flux sampling?

8. About the bioreactor fermentation, please add the OD values and the curves of other metabolites, such as ethanol, glycerol, acetate, pyruvate etc. The curves showed now didn't presented the basic parameters of this newly created Crabtree negative strain. Suggest to read the paper from professor Jack Pronk in Delft.

Reviewer #3

(Remarks to the Author)

The authors have carefully addressed the comments made by all referees on the first version of the manuscript. I still think that there is lack of evidence that the AGT1 gene is different from the MAL11 gene in the *S. cerevisiae* strain used by the authors, but if they could show that the "sucrose null" strain does not grow on sucrose, unless the MAL11 gene is restored, it might be enough for the purpose of the work. Another point is on the description on the maximum specific growth rate calculation, which seems a bit vague to me. For instance, the fact that a minimum of 6 data points in the exponential growth phase was considered, is not mentioned in the new text, only in the answer the authors provided to me. I would suggest they include as much detail as possible on these calculations and, if possible, maybe as a supplementary figure, one or two examples of growth curves, showing the exponential growth phase, highlighting the points that were used for Mimax calculation.

Reviewer #4

(Remarks to the Author)

The authors have responded diligently to my comments and I feel the manuscript could be ready for publication depending on the level of novelty required by this journal and the respective editorial policies.

Version 2:

Reviewer comments:

Reviewer #2

(Remarks to the Author)

The authors have addressed my comments.
It can be accepted.

Reviewer #3

(Remarks to the Author)

I feel a bit concerned with some aspects of this work, which I raised before. Now, after the authors performed additional experiments and responded to the several comments made by the reviewers, it seems that some basics of microbial physiology are missing. For instance, the explanations on the calculation of M_{max} . The authors show a single peak M_{max} value (new supplementary Figure R3), and not a time interval for M_{max} , implying that there is no exponential growth phase in the cultivations performed. This is not what one would expect for a classical yeast growth curve and certainly requires an explanation. Why isn't there a clear exponential growth phase?

Version 3:

Reviewer comments:

Reviewer #3

(Remarks to the Author)

The authors provided an explanation for the calculation of M_{max} values and for the fact that single peak M_{max} values are obtained, using the chosen methodology, instead of constant values during the whole exponential growth phase (EGP), which are normally calculated and observed. They argue that it is a feature of the sliding-window method used and that this does not change the biological interpretation of the results. I am not particularly fond of treating microbial growth data in the way that the authors did since, as I already mentioned, this gives the impression that there is no exponential growth phase, a classical feature of the microbial growth curve, as already pointed out by Jacques Monod in his seminal work (see attached article, Figure 1). To conclude, I believe that the authors should at least give the reader an explanation similar to the one they gave this reviewer, in the sense that growth data can be treated in different ways and that in the way they chose to treat them there are some consequences. However, the real best action to take would be to retreat the growth data using a method that calculates a constant M_{max} value during the EGP.

AUTHOR ACTIONS IN RESPONSE TO REVIEWER COMMENTS

We thank all four reviewers for deep engagement with this work.

Reviewer #1 (Remarks to the Author):

The manuscript introduced the sucrose phosphorylase pathway and deleted the phosphoglucose isomerase gene *PGII*, and decoupling glycolysis from respiration and facilitating the metabolic transition of yeast to a Crabtree negative state. The results are interesting and will be useful for other metabolic engineering research. However, the quality of the manuscript needs to be improved. For example, the function of many enzymes was not described clearly. The result should be analyzed further.

Response:

Thank you for your valuable feedback and suggestions. We are glad that you found our research interesting and potentially impactful. Regarding your comment on the clarity of enzyme functions, we have revised the manuscript to provide clearer descriptions of the enzymes involved, such as LmSP and PvSUF. We also expanded the analysis of our results to provide a more thorough discussion. Below, we address your specific concerns and outline the revisions we have made to improve the quality of the manuscript.

Specific Comments:

Comment #1:

1. Line 109, what reaction do Mal12 and mal32 catalyze?

Response #1:

Thank you for pointing out this detail. MAL12 and MAL32 are key enzymes in the maltose

utilization pathway of *Saccharomyces cerevisiae*. *MAL12* and *MAL32* encodes maltase (α -D-glucosidase), which has the ability to hydrolyze sucrose, maltose, turanose, and other disaccharides. We have clarified this information in the revised manuscript (Line 110-115), and we appreciate the reviewer for bringing this to our attention.

Action #1:

➤ On page 6 of the revised manuscript (Line 110-115), we now write:

“Sucrose metabolism in yeast is governed by several key genes: the extracellular sucrose hydrolysing enzyme gene *SUC2*, the maltase genes *MAL12* and *MAL32*, isomaltase genes *IMA1-5*, and α -glucoside permease genes *MPH2* and *MPH3* (Fig. 1a). These enzymes have the potential to hydrolyse sucrose into glucose and fructose. *SUC2* catalyzes the extracellular hydrolysis of sucrose, whereas the other genes facilitate its intracellular hydrolysis. Additionally, the main sucrose transporters in *S. cerevisiae* are *MAL11* and *MAL31*.”

Comment #2:

2. Line 117, what reaction does LmSP catalyze? Please provide the essential information.

Response #2:

Thank you for your valuable suggestion. In response, we have revised the manuscript to explicitly describe the reaction catalyzed by LmSP. Specifically, LmSP from *Leuconostoc mesenteroides* catalyzes the phosphorolysis of sucrose into fructose and glucose-1-phosphate. This clarification has been incorporated into the revised manuscript (Line 124-126) to improve clarity and ensure a precise understanding of the enzyme's function.

Action #2:

➤ On page 6 of the revised manuscript (Line 124-126), we now write:

“To validate sucrose phosphorolysis, we introduced the *LmSP* gene from *Leuconostoc mesenteroides* into ZQS03, which enables the phosphorolysis of sucrose into fructose and glucose-1-phosphate.”

Comment #3:

3. What is PvSUF? Why the mutant version PvSUF^{I209F C265F G326C} was used? Please at least give a brief introduction of the function of these proteins.

Response #3:

Thank you for your insightful comment. In response, we have revised the manuscript to provide a brief introduction to PvSUF and the rationale for using its mutant version. Specifically, PvSUF, derived from *Phaseolus vulgaris*, facilitates sucrose transport into the cell without ATP consumption (Line 131-133). To address transport constraints, we integrated the mutant version PvSUF^{I209F C265F G326C}, which is believed to exhibit enhanced sucrose transport efficiency compared to the wild-type PvSUF (*Yeast* 2018, 35, 639-652), thereby enabling the strain to regain growth (Line 143-146). This clarification has been incorporated into the revised text to improve readability and ensure a better understanding of the rationale behind our approach.

Action #3:

➤ On page 7 of the revised manuscript (Line 131-133), we now write:

“It has been reported that PvSUF, derived from *Phaseolus vulgaris*, can transport sucrose into the cell without consuming ATP²⁵.”

➤ On page 7 of the revised manuscript (Line 143-146), we now write:

“To circumvent the transport constraint, we integrated the mutant version, PvSUF^{I209F C265F G326C} and LmSP into ZQS05, which is believed to exhibit enhanced sucrose transport efficiency compared to PvSUF, allowing the strain to regain growth³¹.”

Comment #4:

4. In figure 2A, what is ethanol yield rate? Ethanol yield and ethanol production rate are different.

Response #4:

Thank you for your insightful question. We acknowledge that the term "Ethanol yield rate" was an incorrect descriptor. This terminology was originally used to account for ethanol yield of ZQS15 on different carbon sources, as we believed a direct comparison of ethanol production

would not be appropriate. However, we now recognize that the phrasing may have inadvertently caused ambiguity. To address this, the y-axis label has been updated to "Ethanol yield" to ensure precision and improve interpretability.

Action #4:

We have revised the content of Fig. 2a in the manuscript, correcting the previous incorrect description of "Ethanol yield rate" to "Ethanol yield".

Comment #5:

5. It is necessary to measure the NADPH/NADP⁺ level, because pentose phosphate pathway should also be affected.

Response #5:

We would like to thank you for the valuable suggestion. In response, we have measured the NADPH/NADP⁺ ratio in strains ZQS15, ZQS16 (*ΔOCA5*), ZQS17 (*ΔPGII*), and ZQS18 (*ΔOCA5, ΔPGII*) grown in either glucose or sucrose medium (Line 165-171, Line 181-182 and Line 200-204). These measurements have been included in the revised manuscript for clarity and completeness (Supplementary Fig. 4). The results show that the knockout of *OCA5* has minimal impact on the PPP, while *PGII* knockout leads to a reduction in the NADPH/NADP⁺ ratio. Based on these findings, we hypothesize that this reduction is due to the inability of the EMP (glycolytic pathway) to supply the key substrate, glucose-6-phosphate (G6P), required for the PPP following *PGII* deletion. The limitation of G6P results in a reduced NADPH production, leading to the decreased NADPH/NADP⁺ ratio.

We also compared the NADPH/NADP⁺ ratios in three strains related to the synthetic energy system: ZQS39, ZQS48, and ZQS65 (Line 328-333). The results showed that the introduction of the synthetic energy system in ZQS48 and ZQS65 led to a significant decrease in the NADPH/NADP⁺ ratio (Supplementary Fig. 9d). This is due to the action of transhydrogenase (TH), which converts the NADPH produced in the PPP into NADH to balance the NADH/NAD⁺ ratio.

Action #5:

➤ On page 8 of the revised manuscript (Line 165-171), we now write:

“Additionally, we compared the NADH/NAD⁺ and NADPH/NADP⁺ ratios between the two metabolic modes. The reduced NADH/NAD⁺ ratio observed during sucrose phosphorolysis (Fig. 2a) indicates an improved redox balance, partially shifting intracellular metabolism toward a respiration-dependent state. In contrast, the NADPH/NADP⁺ ratio remained unchanged (Supplementary Fig. 4a). Given that the pentose phosphate pathway (PPP) is the primary source of NADPH, this suggests that sucrose phosphorolysis does not affect PPP metabolism.”

➤ On page 9 of the revised manuscript (Line 181-182), we now write:

“Meanwhile, the NADPH/NADP⁺ ratio remained unchanged (Supplementary Fig. 4e), suggesting that the deletion of *OCA5* does not affect the PPP.”

➤ On page 10 of the revised manuscript (Line 200-204), we now write:

“Additionally, the NADPH/NADP⁺ ratio in ZQS17 and ZQS18 was lower compared to ZQS15 and ZQS16 (Supplementary Fig. 4f), indicating that the knockout of *PGII* reduces the PPP flux. This is due to the inability of the EMP to supply glucose-6-phosphate (G6P), a key substrate for the PPP, following *PGII* deletion. The limitation of G6P results in a reduced NADPH production, thereby lowering the NADPH/NADP⁺ ratio.”

➤ On page 15 of the revised manuscript (Line 328-333), we now write:

“Compared to ZQS39, ZQS65 maintains a lower NADPH/NADP⁺ ratio (Supplementary Fig. 11d) while keeping a constant NADH/NAD⁺ ratio, indicating that the synthetic energy system efficiently converts NADPH into NADH to sustain cytosolic NADH/NAD⁺ balance. Additionally, the higher ATP content in ZQS65 (Fig. 5c) suggests that the modified synthetic energy system enables yeast to bypass fermentation, accelerating NADH flux into the respiratory chain and relying solely on oxidative phosphorylation for energy production.”

Supplementary Fig. 4. Validation of central carbon metabolism-related parameters in strains ZQS15-ZQS18.

a. NADPH/NADP⁺ ratio of strain ZQS15 under glucose metabolism and sucrose phosphorolysis metabolic modes. **b.** Ethanol accumulation in ZQS16 during glucose metabolism. **c.** Glycerol accumulation in ZQS16 during glucose metabolism. **d.** Comparison of the NADH/NAD⁺ ratio between ZQS16 and ZQS15 during glucose metabolism. **e.** NADPH/NADP⁺ ratio of strain ZQS16 under sucrose phosphorolysis metabolic mode. **f.** NADPH/NADP⁺ ratio in strains ZQS17 and ZQS18. All data are presented as mean ± SD of biological triplicates. Statistical analysis was conducted using Student's *t*-test (two-sample, two-tailed; **P* value < 0.05, ****P* value < 0.001; NS represents no significant difference; sample size, *n* = 3).

Supplementary Fig. 11. Impact of constructing and optimizing synthetic energy systems on cell growth, NADH/NAD⁺ and NADPH/NADP⁺ ratio.

d. NADH/NAD⁺ and NADPH/NADP⁺ ratio in strains with the synthetic energy system. ZQS39 is the control strain without the synthetic energy system, ZQS48 contains the synthetic energy system, and ZQS65 contains the optimized synthetic energy system.

Comment #6:

6. The products mentioned in the manuscript such as lactate, 3-HP, *p*-coumaric acid and farnesene, although the production of these chemical is improved, I think the reason of the improvement are different. Please discuss them in detail. For example, lactate and ethanol share the same precursor pyruvate, however, 3-HP and farnesene are derived from acetyl-CoA. Differently, *p*-coumaric acid requires E4P and PEP.

Response #6:

Thank you very much for your valuable comments and constructive feedback. We appreciate

your suggestion to discuss the different reasons behind the improvements in the production of lactate, 3-HP, *p*-coumaric acid, and farnesene, as each product is derived from distinct metabolic pathways.

In response to your comment, we have revised the manuscript to include a detailed discussion of these four products and their respective biosynthetic pathways (Line 355-361). Meanwhile, we clarified the role of *p*-coumaric acid and farnesene as representative products of the shikimate and mevalonate (MVA) pathways, respectively. The engineered strain ZQS65 was specifically selected to evaluate the fluxes of pyruvate and acetyl-CoA supply, as well as its ability to efficiently channel carbon flux through the shikimate and MVA pathways for product synthesis. We divided it into four key points to overall evaluate the potential of ZQS65 for different products.

Additionally, during the product validation process of the engineered strains, we added conclusions to facilitate the analysis of the strains' potential in synthesizing different products (Line 369-372, Line 380-384 and Line 395-398). We believe these revisions address your concern by providing a clearer understanding of the underlying metabolic pathways and engineering strategies used to improve the production of these chemicals.

Action #6:

➤ On page 16 of the revised manuscript (Line 355-361), we now write:

“Lactate is derived from pyruvate, while 3-HP originates from acetyl-CoA (Fig. 6a). *p*-coumaric acid and farnesene serve as representative products of the shikimate and mevalonate (MVA) pathways, respectively (Fig. 6a). These four products were selected to evaluate the metabolic flux distribution in strain ZQS39 and ZQS65, specifically assessing its capacity for pyruvate and acetyl-CoA supply, as well as its efficiency in channeling carbon flux toward the shikimate and MVA pathways for product synthesis.”

➤ On page 16-17 of the revised manuscript (Line 369-372), we now write:

“However, the lactate synthesis ability of ZQS68 is lower than that of ZQS67 (Fig. 6c), which lacks the synthetic energy system. This indicated that the introduction of the synthetic energy system consumes cytosolic NADH, thereby competing with lactate synthesis.”

➤ On page 17 of the revised manuscript (Line 380-384), we now write:

“The engineered strain ZQS71 produced 3-HP at a titer of 1.96 g/L with a yield of 0.15 g/g sucrose, which is 8.2-fold higher than that of ZQS69 and 3.1-fold improvement compared to ZQS70 (Fig. 6e). This result indicated that Crabtree-negative strains equipped with the synthetic energy system exhibit an enhanced supply of acetyl-CoA for the synthesis of their derivatives.”

➤ On page 18 of the revised manuscript (Line 395-398), we now write:

“The result indicated that the engineered strain retains a competitive edge in the shikimate pathway. However, we hypothesize that the suboptimal yield enhancement of *p*-coumaric acid, compared to 3-HP, could be attributed to its more extended biosynthetic route, which may introduce inefficiencies such as metabolic flux dilution.”

Comment #7:

7. ZQS39 does not accumulated any ethanol, and the authors further engineered the strain to generate more NADH and more ATP. Are the engineered strains better than ZQS39? Asides from ATP generation, which parameters are better?

Response #7:

Thank you for your insightful comments and for raising the important question regarding the comparison of the engineered strains to ZQS39. We appreciate your suggestion to assess parameters beyond just ATP generation.

After obtaining strain ZQS39, we further engineered it by introducing the synthetic energy system to utilize intracellular NADH for enhanced ATP generation. This modification led to the creation of strain ZQS65, which enables a greater flow of NADH into the electron transport chain for ATP synthesis. In addition to evaluating ATP generation, we compared the NADH/NAD⁺ and NADPH/NADP⁺ ratios between ZQS39 and ZQS65 (Line 328-333). The results showed that the NADH/NAD⁺ ratio in ZQS65 remained stable, while the NADPH/NADP⁺ ratio significantly decreased (Supplementary Fig. 9d). This suggests that the synthetic energy system in ZQS65 effectively converts NADPH into NADH, which is subsequently utilized for ATP production. This also provides further validation of the functionality of the synthetic energy system.

To assess whether ZQS65 performs better than ZQS39, we introduced the biosynthetic pathways for the four target products (lactate, 3-HP, *p*-coumaric acid and farnesene) into ZQS39, generating strains ZQS67, ZQS70, ZQS73, and ZQS76 (Fig. 6). These strains were then compared with corresponding strains derived from ZQS65. The results showed that ZQS39 outperformed ZQS65 in lactate production (Fig. 6c), while ZQS65 exhibited superior performance in the synthesis of 3-HP, *p*-coumaric acid, and farnesene (Fig. 6e, g, i). We observed that lactate synthesis relies on NADH supply, and since the synthetic energy system in ZQS65 uses NADH for ATP production, lactate synthesis was reduced in ZQS65. However, the synthetic energy system significantly boosted 3-HP production, confirming the benefits of NADH-driven ATP generation in enhancing certain biosynthetic pathways.

In conclusion, ZQS65 outperforms ZQS39 in terms of the production of 3-HP, *p*-coumaric acid, and farnesene, while ZQS39 is more efficient in lactate production. The engineered energy synthesis system in ZQS65 optimizes NADH utilization for ATP generation, which improves the metabolic capacity for certain product syntheses.

Action #7:

Supplementary Fig. 11. Impact of constructing and optimizing synthetic energy systems on cell growth, NADH/NAD⁺ and NADPH/NADP⁺ ratio.

d. NADH/NAD⁺ and NADPH/NADP⁺ ratio in strains with the synthetic energy system. ZQS39 is the control strain without the synthetic energy system, ZQS48 contains the synthetic energy system, and ZQS65 contains the optimized synthetic energy system.

Fig. 6. The Crabtree-negative strain with the synthetic energy system was applied to the

synthesis of compounds through different pathways.

Comment #8:

8. Please explain why in this engineered strain, the carbon can fully enter the TCA cycle and respiratory chain for energy metabolism.

Response #8:

Thank you for raising this important point. In wild-type *Saccharomyces cerevisiae*, carbon flux is dynamically regulated by central carbon metabolism to maintain cellular functions. Wild-type Crabtree-positive strains prioritize ethanol production under high-glucose conditions due to the Crabtree effect. Although ethanol can be partially re-assimilated in later stages, residual ethanol persists at the end of fermentation, leading to carbon loss. In addition to carbon loss, due to the Crabtree effect, yeast preferentially metabolizes glucose through fermentation under aerobic conditions, resulting in the production of only 2 ATP molecules per glucose molecule, whereas respiration can generate approximately 32 ATP molecules. In contrast, our engineered Crabtree-negative strain eliminates ethanol overflow metabolism. This genetic modification redirects glycolytic flux (via the EMP pathway) toward mitochondrial oxidation: pyruvate is converted to acetyl-CoA by the pyruvate dehydrogenase complex (PDH), which fully enters the TCA cycle.

The TCA cycle is tightly coupled with the respiratory chain for ATP synthesis. In mitochondria, NADH generated from the TCA cycle donates electrons to the electron transport chain (ETC), ultimately reducing oxygen to water and driving oxidative phosphorylation. To further enhance energy efficiency, we introduced the synthetic energy system (mitochondrial outer membrane NADH dehydrogenases NDE1, NDE2 and transhydrogenase AcTH) to harness cytosolic NADH and NADPH for ATP production. This strategy alleviates redox imbalance and ensures maximal carbon flux toward the TCA cycle and respiratory chain. In summary, the elimination of ethanol overflow metabolism in our Crabtree-negative strain, combined with the introduction of the synthetic energy system, enables carbon atoms to fully enter the TCA cycle and respiratory chain, thereby optimizing energy metabolism.

We hope this explanation clarifies the metabolic rewiring in our engineered strain. Please let

us know if further details or clarifications are needed. We greatly value your feedback and are committed to improving the manuscript based on your suggestions.

Reviewer #2 (Remarks to the Author):

In this manuscript, the authors engineered the sucrose phosphorolysis and EMP pathway to restrict the importing sugar into the cell, coupling the reported inositol pyrophosphatase, a Crabtree-negative yeast was created alongside an obvious lag phase. Then reverse engineering was employed based on the transcriptional analysis, which shortened the lag phase to some extent. Furtherly, the redox equivalent balance between cytosolic and mitochondria was engineered, leading to ethanol and glycerol overflow and weaken cell growth. Efficient producing the non-ethanol chemicals demonstrated the advantage of this newly constructed yeast. This paper employed to a new strategy to shift *S. cerevisiae* from Crabtree positive to negative. Compared with the published paper, the novelty is not strong enough. The main reason is that the logic behind this strategy is the same as previous one: restricting sugar import and glycolysis to adjust the balance of redox equivalent between fermentation and respiration. In the other word, new molecular mechanism and knowledge was lacking, which is not supportive for publishing this work in Nat Comm at this stage.

Response:

We sincerely appreciate your thorough review and valuable feedback on our manuscript. We understand your concern regarding the novelty of our work and the underlying strategy of restricting sugar import and glycolysis to rebalance redox equivalents. Below, we address your comments in detail and clarify the distinct contributions of our study.

Our work presents a novel and integrated strategy for overcoming the Crabtree effect in *Saccharomyces cerevisiae*. While previous studies have focused on limiting glycolysis to promote respiratory metabolism, our approach differs in the following key aspects:

1. Sucrose phosphorolysis for efficient carbon utilization: Instead of directly restricting glucose uptake, we introduced a sucrose phosphorolysis pathway, which enables energy-efficient sucrose utilization by bypassing ATP-consuming hexokinase activity. This approach maximizes carbon conservation and enhances metabolic efficiency.
2. Metabolic reprogramming through *PGII* deletion: The deletion of *PGII* fundamentally restructures central carbon metabolism by blocking the entry of glucose-6-phosphate into glycolysis, thereby shifting metabolic flux toward respiration. Unlike previous strategies that

rely on glucose transport regulation, our approach directly reconfigures metabolic network topology to induce a Crabtree-negative phenotype.

3. Synthetic energy system for NADH/NAD⁺ regulation: To address the redox imbalance resulting from metabolic rewiring, we engineered a synthetic energy system that efficiently channels cytosolic NADH into ATP production. This not only mitigates redox stress but also ensures a sufficient ATP supply for cell growth and biosynthesis.

4. Demonstration of broad applicability in bioproduction: Our engineered strain exhibited significantly enhanced yields of multiple non-ethanol chemicals, including lactic acid and 3-hydroxypropionic acid (8- to 11-fold increases), as well as *p*-coumaric acid and farnesene. These improvements highlight the broad applicability of our strategy for metabolic engineering in yeast.

In response to your comments, we have strengthened our manuscript on the novelty of our approach and included a more detailed comparison with previous studies. We hope these clarifications address your concerns and demonstrate the unique contributions of our work. Thank you again for your valuable suggestions, which have helped us refine our manuscript. We appreciate your time and effort in reviewing our study.

Specific Comments:

Comment #1:

1. The specific growth rates of the engineered strains, especially the key one, are needed to be presented in this work. This parameter is quite important for describing the Crabtree effect related studies and comparison with reported works.

Response #1:

Thank you very much for your careful review and valuable comments on our manuscript. We greatly appreciate your suggestion regarding the specific growth rates of the engineered strains. In response to your suggestion, we have added the specific growth rate data throughout the manuscript. Specifically, we have included the specific growth rate data for all engineered strains, particularly the key strain, in the Results section. Additionally, we have provided a detailed description of the measurement method for specific growth rates in the Methods

section (Line 574-580). We believe these additions have strengthened the manuscript by providing essential parameters for evaluating the Crabtree effect and facilitating comparisons with other studies. The specific growth rate data further support our conclusions and enhance the overall quality of our work.

Once again, we sincerely appreciate your constructive feedback, which has helped us improve the manuscript. If you have any further concerns or suggestions regarding the revised content, we would be happy to address them.

Action #1:

➤ On page 6-7 of the revised manuscript (Line 126-128), we now write:

“As a result, strain ZQS04 restored sucrose metabolism and exhibited a higher maximum specific growth rate (μ_{\max}) of 0.20 h⁻¹ compared to the parental strain C800 with a μ_{\max} of 0.13 h⁻¹ (Fig. 1d and Supplementary Fig. 2b).”

➤ On page 7 of the revised manuscript (Line 140-142), we now write:

“Strain ZQS13 with replenished *MAL11* expression recovered growth in the sucrose medium with a μ_{\max} of 0.20 h⁻¹ (Fig. 1f and Supplementary Fig. 2b).”

➤ On page 7 of the revised manuscript (Line 147-149), we now write:

“Therefore, we co-expressed the mutated *PvSUF*^{I209F C265F G326C} and *MAL11* resulted in strain ZQS15, which achieved a μ_{\max} of 0.21 h⁻¹ (Fig. 1g and Supplementary Fig. 2b).”

➤ On page 9 of the revised manuscript (Line 176-178), we now write:

“Consequently, we knocked out the *OCA5* gene in strain ZQS15 to generate strain ZQS16, which exhibited a similar growth profile to ZQS15 (Fig. 2d), with a μ_{\max} of 0.21 h⁻¹ (Supplementary Fig. 3a).”

➤ On page 9-10 of the revised manuscript (Line 193-199), we now write:

“Notably, although the μ_{\max} of the *PGII* knockout strains ZQS17 and ZQS18 were similar to that of ZQS16 (Supplementary Fig. 3a), both exhibited a longer lag phase time compared to ZQS16 (Supplementary Fig. 3b). We speculate that the knockout of *PGII* reduces the EMP pathway flux, thereby extending the cell growth cycle. Moreover, we found that the lag phase time of ZQS17 reached 34.1 h, significantly longer than that of ZQS18 with 23.5 h (Supplementary Fig. 3b), indicating that *OCA5* deletion improves the growth performance of

Crabtree-negative yeast strains.”

➤ On page 11 of the revised manuscript (Line 238-240), we now write:

“ZQS16/ZQS15 strains exhibited higher μ_{\max} on sucrose than on glucose, while ZQS18/ZQS17 (*PGII* Δ) showed a longer lag phase time on sucrose than ZQS16/ZQS15 (Supplementary Fig. 8).”

➤ On page 15 of the revised manuscript (Line 326-327), we now write:

“In strain ZQS65, ethanol and glycerol levels were reduced to minimal (Fig. 5g, h) without compromising growth, with a μ_{\max} of 0.20 h⁻¹ (Supplementary Fig. 11b, c).”

The figures illustrating the measurement of specific growth rates are presented in Supplementary Fig. 2b, Supplementary Fig. 3a, Supplementary Fig. 8b, Supplementary Fig. 11c and Supplementary Fig. 12c, respectively.

Comment #2:

2. Since this new yeast showed superiority on producing non-ethanol chemicals, batch fermentations of the key strains such as ZQS18 and ZQS65, etc. in bioreactor to characterize the specific growth rate, biomass yield, sugar uptake rate are suggested. This would be helpful for the community to get to know this strain for further study.

Response #2:

We sincerely appreciate your insightful suggestion regarding the characterization of the fermentation performance of the key strains, ZQS18 and ZQS65. In response to your suggestion, we have conducted comprehensive batch fermentation experiments for both ZQS65 and ZQS18 in a 5-L fermenter. The results revealed that ZQS65, equipped with the synthetic energy system, demonstrated superior growth performance compared to ZQS18. Specifically, ZQS65 achieved a μ_{\max} of 0.24 h⁻¹, a maximum biomass yield of 0.24 g DCW/g sucrose, and a sucrose uptake rate of 0.17 g/g DCW/h. Interestingly, while both strains reached maximum biomass at 48 h, ZQS65 continued to utilize sucrose until it was completely depleted. During this process, neither strain exhibited accumulation of ethanol or glycerol. This suggests that ZQS65 exhibits enhanced metabolic activity during the growth stagnation phase, likely due to the synthetic energy system's ability to continuously supply energy. These findings,

presented in Supplementary Fig. 13 and discussed in the revised manuscript (Line 333-345).

Once again, we sincerely appreciate your constructive suggestion, which has significantly improved the quality of our work.

Action #2:

➤ On page 15-16 of the revised manuscript (Line 333-345), we now write:

“To further validate the fermentation performance of strain ZQS65, we conducted batch fermentation in a 5-L fermenter for both ZQS65 and ZQS18. The results demonstrated that ZQS65, incorporating the synthetic energy system, achieved a μ_{\max} of 0.24 h^{-1} , a maximum biomass yield of $0.24 \text{ g DCW/g sucrose}$, and a sucrose uptake rate of 0.17 g/g DCW/h (Supplementary Fig. 13). In comparison, ZQS18 exhibited slightly lower values (0.22 h^{-1} , $0.24 \text{ g DCW/g sucrose}$, 0.16 g/g DCW/h), highlighting the superior growth performance of ZQS65. Notably, while ZQS18 ceased sucrose consumption after reaching maximum biomass at 48 h, ZQS65 continued to gradually utilize sucrose until it was completely depleted, even after reaching maximum biomass at the same time point (Supplementary Fig. 13). This observation suggests that ZQS65 could efficiently utilize carbon sources and maintain high metabolic activity during the growth stagnation phase, likely due to the synthetic energy system's ability to continuously support cellular energy demands.”

Supplementary Fig. 13. The growth status of ZQS65 and ZQS18 in a 5L-fermenter.

The vertical axes represent DCW (indicating dry cell weight) and Residual sucrose (indicating

the residual sucrose concentration in the fermentation broth), respectively. The initial sucrose concentration was 20 g/L.

Comment #3:

3. How about the growth profiles of strains from 59-65?

Response #3:

Thank you very much for your careful review and constructive comments on our manuscript. We greatly appreciate your question regarding the growth profiles of strains 59-65.

In response to your question, we have included the growth curves for these strains in Supplementary Fig. 11b of the revised manuscript. Additionally, to provide a more intuitive understanding of the growth dynamics, we have calculated and included the specific growth rates for these strains in the Supplementary Fig. 11c. These data collectively offer a comprehensive view of the growth behavior of the strains in question. The results indicate that the knockout of genes related to overflow metabolism in strain ZQS59 had a minor impact on cell growth, with the final strain ZQS65 exhibiting a μ_{\max} of 0.20 h⁻¹.

We hope that these additions address your concern and provide the necessary insights into the growth profiles of the strains. Please let us know if there are any further details or clarifications you would like us to provide. Thank you again for your valuable feedback, which has greatly improved the quality of our work.

Action #3:

Supplementary Fig. 11. Impact of constructing and optimizing synthetic energy systems on cell growth, NADH/NAD⁺ and NADPH/NADP⁺ ratio.

b. Growth curves of strains ZQS59-ZQS65. **c.** The specific growth rate of strains ZQS59-ZQS65.

Comment #4:

4. About the lag phase, some of genes showed positive role, what is the explanation? More discussion is needed.

Response #4:

Thank you for your constructive feedback on our manuscript. We appreciate your suggestion to further discuss the genes that showed a positive role in reducing the lag phase. In response to your comment, we have expanded the discussion to address the possible explanations for the observed effects and the underlying mechanisms.

In the revised manuscript (Line 272-285), we explain that the significant reduction in the lag phase observed in ZQS39 (*RGI2* Δ) can be attributed to several factors related to central carbon metabolism, as shown in Supplementary Fig. 10. Comparative analysis between ZQS39

(*RGI2*Δ) and ZQS18 revealed that the NADH/NAD⁺ ratio in ZQS39 is lower, suggesting an enhanced redox balance that could facilitate more efficient glycolytic flux and quicker metabolic adaptation (Supplementary Fig. 10a). Importantly, the NADPH/NADP⁺ ratio remained unchanged (Supplementary Fig. 10b), indicating that the pentose phosphate pathway was not significantly affected by the deletion of *RGI2*. Furthermore, ATP content in ZQS39 increased (Supplementary Fig. 10c), supporting the hypothesis that *RGI2* deletion promotes glycolysis over respiration, accelerating energy production and facilitating a more rapid transition from lag to exponential growth.

These findings indicate that *RGI2* influences the metabolic balance between glycolysis and the TCA cycle, reducing competition between these pathways and promoting a more efficient metabolic transition. We have further discussed the role of these metabolic changes in shortening the lag phase and improving growth kinetics. We hope this added discussion provides a more comprehensive explanation for the observed effects.

Thank you again for your insightful comments, which have helped us clarify and strengthen this aspect of our study.

Action #4:

➤ On page 13 of the revised manuscript (Line 272-285), we now write:

“The significant reduction in the lag phase observed in ZQS39 (*RGI2* Δ) suggests that *RGI2* plays a crucial role in regulating metabolic adaptation during the early growth phase. This effect may be attributed to several factors associated with central carbon metabolism (Supplementary Fig. 10). A comparative analysis between ZQS39 (*RGI2*Δ) and ZQS18 revealed that the NADH/NAD⁺ ratio in ZQS39 was lower (Supplementary Fig. 10a), suggesting an enhanced redox state. Notably, the NADPH/NADP⁺ ratio remained unchanged (Supplementary Fig. 10b), which implies that the PPP was not significantly affected by *RGI2* deletion. In parallel, the ATP content in ZQS39 increased (Supplementary Fig. 10c), supporting the hypothesis that *RGI2* deletion accelerates energy production by favoring glycolysis over respiration. These observations imply that *RGI2* influences the metabolic balance between glycolysis and the TCA cycle, potentially reducing competition between these pathways. Overall, these findings indicate that *RGI2* deletion facilitates a more efficient metabolic

transition, ultimately contributing to the observed reduction in lag phase duration.”

Supplementary Fig. 10. Comparison of central carbon metabolism-related parameters between ZQS39 and ZQS18.

ZQS18 is the control strain, and ZQS39 represents the *RGI2* knockout strain. **a.** Comparison of the NADH/NAD⁺ ratio between ZQS39 and ZQS18. **b.** Comparison of the NADPH/NADP⁺ ratio between ZQS39 and ZQS18. **c.** Comparison of ATP content between ZQS39 and ZQS18. All data are presented as mean ± SD of biological triplicates. Statistical analysis was conducted using Student’s *t*-test (two-sample, two-tailed; **P* value < 0.05, ****P* value < 0.001; NS represents no significant difference; sample size, *n* = 3).

Comment #5:

5. About the title, I don’t think the carbon redistribution was driven by sucrose. And carbon redistribution may not be the main reason for eliminating Crabtree effect but the redox equivalent may play the essential role.

Response #5:

Thank you for your insightful comments on our manuscript. We appreciate your critical evaluation, which has helped us refine our study.

Regarding your concern about the title, we acknowledge that carbon redistribution may not be the primary mechanism driving the elimination of the Crabtree effect. Instead, our results suggest that redox balance plays a crucial role in facilitating this metabolic shift. In response

to your feedback, we have revised the title to:

“Sucrose-driven carbon redox rebalancing eliminates the Crabtree effect and boosts energy metabolism in yeast.”

This revised title more accurately reflects the key findings of our study, emphasizing the role of sucrose metabolism and redox balance in overcoming the Crabtree effect while de-emphasizing carbon redistribution as the primary mechanism. We believe this change improves clarity and aligns with the core message of our work.

Thank you again for your valuable suggestions, and we hope this revision addresses your concerns. Please let us know if you have any further recommendations.

Reviewer #3 (Remarks to the Author):

The manuscript describes a massive amount of work, mainly involving recombinant DNA and the overexpression/deletion of several genes in the yeast *Saccharomyces cerevisiae*. The metabolic engineering strategies used had in part already been published by other groups before, but some seem to be new. At the end of the day, the authors claim to have generated a platform strain to produce different chemicals, which metabolizes sucrose mainly via respiration, leading to higher ATP yields and carbon availability, when compared to strains that preferably metabolize sugars via ethanolic fermentation. This, per se, can be considered a very interesting result.

There are some technical and conceptual details that could be addressed here, but I would like to mention three aspects that need clarification, before we can be sure that the work deserves publication as it is:

Response:

Thank you for your detailed review of our manuscript and for the positive evaluation of our work. We are delighted to hear that you found our platform strain and its ability to metabolize sucrose predominantly via respiration, resulting in increased ATP yields and carbon availability, to be interesting and meaningful. Your recognition greatly encourages us and reinforces the value of our research direction.

We also sincerely appreciate your constructive feedback highlighting technical and conceptual aspects that require clarification. We fully understand the importance of addressing these points to enhance the quality of our manuscript. In the revised version, we have carefully addressed each of the three aspects you mentioned and made the necessary revisions to provide comprehensive clarifications.

Specific Comments:**Comment #1:**

All comparisons made between the different strains, in terms of growth, are mainly based on the presentation of OD versus time graphs. I could not find calculations of specific growth rates, except on suppl. fig. 7. There is no indication on how these specific growth rates were

calculated and I doubt whether one can perform good calculations, considering the small number of experimental points during the exponential growth phase in the growth curves presented throughout the work. I believe that a rigorous calculation of specific growth rates is important to compare the growth of different strains;

Response #1:

Thank you for your constructive feedback regarding the calculation of specific growth rates and the number of experimental points during the exponential growth phase. In response to your concerns, we have carefully reanalyzed the growth curves of all relevant strains mentioned in the manuscript. To address the issue, we ensured that at least six data points were collected during the exponential growth phase for each growth curve. This additional data guarantees the accuracy and reliability of the specific growth rate calculations. Moreover, we have updated the Methods (Line 574-580) section to include a detailed description of how specific growth rates were calculated to improve transparency and reproducibility. We believe these revisions address your concerns and enhance the robustness of our findings.

Action #1:

➤ On page 6-7 of the revised manuscript (Line 126-128), we now write:

“As a result, strain ZQS04 restored sucrose metabolism and exhibited a higher maximum specific growth rate (μ_{\max}) of 0.20 h⁻¹ compared to the parental strain C800 with a μ_{\max} of 0.13 h⁻¹ (Fig. 1d and Supplementary Fig. 2b).”

➤ On page 7 of the revised manuscript (Line 140-142), we now write:

“Strain ZQS13 with replenished *MAL11* expression recovered growth in the sucrose medium with a μ_{\max} of 0.20 h⁻¹ (Fig. 1f and Supplementary Fig. 2b).”

➤ On page 7 of the revised manuscript (Line 147-149), we now write:

“Therefore, we co-expressed the mutated *PvSUF^{I209F C265F G326C}* and *MAL11* resulted in strain ZQS15, which achieved a μ_{\max} of 0.21 h⁻¹ (Fig. 1g and Supplementary Fig. 2b).”

➤ On page 9 of the revised manuscript (Line 176-178), we now write:

“Consequently, we knocked out the *OCA5* gene in strain ZQS15 to generate strain ZQS16, which exhibited a similar growth profile to ZQS15 (Fig. 2d), with a μ_{\max} of 0.21 h⁻¹ (Supplementary Fig. 3a).”

➤ On page 9-10 of the revised manuscript (Line 193-199), we now write:

“Notably, although the μ_{\max} of the *PGII* knockout strains ZQS17 and ZQS18 were similar to that of ZQS16 (Supplementary Fig. 3a), both exhibited a longer lag phase time compared to ZQS16 (Supplementary Fig. 3b). We speculated that the knockout of *PGII* reduced the EMP pathway flux, thereby extending the cell growth cycle. Moreover, we found that the lag phase time of ZQS17 reached 34.1 h, significantly longer than that of ZQS18 with 23.5 h (Supplementary Fig. 3b), indicating that *OCA5* deletion improves the growth performance of Crabtree-negative yeast strains.”

➤ On page 11 of the revised manuscript (Line 238-240), we now write:

“ZQS16/ZQS15 strains exhibited higher μ_{\max} on sucrose than on glucose, while ZQS18/ZQS17 (*PGII* Δ) showed a longer lag phase time on sucrose than ZQS16/ZQS15 (Supplementary Fig. 8).”

➤ On page 15 of the revised manuscript (Line 326-327), we now write:

“In strain ZQS65, ethanol and glycerol levels were reduced to minimal (Fig. 5g, h) without compromising growth, with a μ_{\max} of 0.20 h⁻¹ (Supplementary Fig. 11b, c).”

The figures illustrating the measurement of specific growth rates are presented in Supplementary Fig. 2b, Supplementary Fig. 3a, Supplementary Fig. 8b, Supplementary Fig. 11c and Supplementary Fig. 12c, respectively.

Comment #2:

There is no mentioning of the *AGTI* gene in the work. This is an important alpha-glucoside transporter that can transport sucrose. This gene is different from *MAL11*, although no distinction is made by the yeastgenome.org database. According to some colleague researchers, this is an error, which has been reported to SGD, but was never corrected. Here are two articles in which this distinction appears: Cousseau, F. E., Alves, S. L., Jr, Trichez, D., & Stambuk, B. U. (2013). Characterization of maltotriose transporters from the *Saccharomyces eubayanus* subgenome of the hybrid *Saccharomyces pastorianus* lager brewing yeast strain Weihenstephan 34/70. *Letters in applied microbiology*, 56(1), 21–29. <https://doi.org/10.1111/lam.12011>. AND Trichez, D., Knychala, M. M., Figueiredo, C. M.,

Alves, S. L., Jr, da Silva, M. A., Miletti, L. C., de Araujo, P. S., & Stambuk, B. U. (2019). Key amino acid residues of the *AGTI* permease required for maltotriose consumption and fermentation by *Saccharomyces cerevisiae*. *Journal of applied microbiology*, 126(2), 580–594. <https://doi.org/10.1111/jam.14161>

According to this expert with whom I discussed the issue, "*AGTI* and *MAL11* are different genes! The confusion comes from the "yeastgenomedatabase" which confused everything. As *AGTI* is on chromosome VII, where the *MAL1* locus is located, they mistakenly call it "*MAL11*". To <name hidden> (who cloned *AGTI* for the first time) has already complained about it, I have also complained about it, but without success..... The *MAL11* gene/transporter has been extensively studied by *Rosario Lagunas*, she even cloned and deposited the gene sequence in GenBank..... "

"in GenBank there are several entries for *AGTI*, one of them is CAA97322. The one for *MAL11* is CAA10168. You will see that the sequences are different (I think 57% identity), and while *MAL11* aligns with *MAL31*, *MAL61* and *MAL21* (the other transporters from the classic *MAL* loci), *AGTI* forms a separate group"

It is important to understand whether the authors have paid attention to this or not, since the deletion of both *MAL11* and *AGTI* is presumably necessary to eliminate sucrose transport in *S. cerevisiae*.

Response #2:

Thank you for your insightful question. The *AGTI* gene you mentioned is an aspect we had not considered. Therefore, we compared the genes CAA97322 (*AGTI*) and CAA10168 (*MAL11*) from the NCBI database (*Journal of Applied Microbiology.*, 2018, 126(2), 580 – 594). However, we found that the amino acid sequence of CAA97322 (*AGTI*) is identical to that of the *MAL11* gene in the *Saccharomyces cerevisiae* Genome Database (SGD). Meanwhile, CAA10168 (*MAL11*) shares homology with the *MAL31* gene in the SGD, differing by only four amino acids (Fig. R1a). Additionally, our comparison of amino acid sequences revealed that *MAL11* (CAA10168) and *MAL31* (O93979) are identical according to the reference (*Journal of Applied Microbiology.*, 2018, 126(2), 580 – 594). Based on the above, we conclude that the *AGTI* gene you referred to corresponds to the *MAL11* gene in the SGD, and the *MAL11* gene in the SGD

corresponds to *MAL31*.

In light of this conclusion, we revisited our manuscript. Regarding *MAL11*, we compared the growth profiles of the strain ZQS06 (Δ *MAL11*) and the *MAL11* overexpression strain ZQS13 (Fig. 1d and Fig. 1f). The results demonstrated that sucrose transport was restored only when *MAL11* was overexpressed. This finding confirms that *MAL11* is the primary sucrose transporter. Regarding *MAL31*, we identified an error in the original manuscript and have corrected it. During the deletion of sucrose hydrolysis-related genes, *MAL31* was deleted together with *MAL32* (Fig. R1b), indicating that the genotype of ZQS03 and its derived strains is *suc2* Δ ; *ima1* Δ ; *ima2* Δ ; *ima3* Δ ; *ima4* Δ ; *ima5* Δ ; *mph2* Δ ; *mph3* Δ ; *mal12* Δ ; *mal31-mal32* Δ . To further verify the sucrose transport ability of *MAL31*, we overexpressed the *MAL31* gene in strain ZQS06. However, the strain ZQS06+*MAL31* was still unable to utilize sucrose (Fig. R1c). Based on these results, we conclude that *MAL11*, rather than *MAL31*, is the primary sucrose transporter.

Based on the information we have gathered so far, these are the conclusions we have reached. However, if you have more accurate information or believe that my conclusions are incorrect, we would be deeply grateful if you could share your insights with us. We truly value your expertise and are fully committed to working together to ensure the accuracy and integrity of this work. Your continued guidance would be immensely appreciated as we strive to improve this study.

Action #2:

The genotypes of the relevant strains have been updated in Supplementary Table S1. Strains. We have generated relevant figures to further support the results (Fig. R1). This figure is primarily intended to explain experimental details and have not been included in the main manuscript or supplementary materials to avoid redundancy in the main text.

Fig. R1. Supporting evidence for the gene *MAL31*.

a. Alignment of *MAL31* (from SGD) with CAA10168 (annotated as *MAL11* in NCBI), showing only four amino acid differences. **b.** PCR validation of whether *MAL31* was co-deleted with *MAL32* in the strain ZQS03. The PCR product of *MAL31-MAL32*-intact strain C800 was 4973 bp, while that of the Δ *MAL31-MAL32* strain ZQS03 was 498 bp. **c.** Growth profiles of the *MAL31* overexpression strain.

Comment #3:

3. Some important parameters displayed in the Figures are not clear, e.g. the "Ethanol yield rate" in Figure 2a. I understand what an Ethanol yield or a specific ethanol production rate are, but it is not clear to me what an Ethanol yield rate is.

Response #3:

Thank you for your insightful question. We acknowledge that the term "Ethanol yield rate" was an incorrect expression. This terminology was originally used to account for ethanol yield of ZQS15 on different carbon sources, as we believed a direct comparison of ethanol production would not be appropriate. However, we now recognize that this may have caused confusion. To address this, we have replaced the y-axis label with "Ethanol yield" in the revised figure for clarity and accuracy.

Action #3:

We have revised the content of Fig. 2a in the manuscript, correcting the previous incorrect description of "Ethanol yield rate" to "Ethanol yield".

Reviewer #4 (Remarks to the Author):

The manuscript describes the development of a Crabtree-negative yeast with enhanced energy system and its applications. The authors used sucrose as the carbon source and introduced sucrose phosphorylase to eliminate glucose repression and improve energy conservation. The authors conducted a comprehensive study to alleviate the Crabtree effect and boost energy production. The following comments may help to improve the paper:

Response:

We sincerely appreciate your thorough evaluation of our manuscript and your positive feedback regarding our study on the development of a Crabtree-negative yeast with an enhanced energy system. We are grateful for your recognition of our efforts in employing sucrose as a carbon source, introducing sucrose phosphorylase to mitigate glucose repression, and improving energy conservation.

Your detailed comments are invaluable in improving the quality and clarity of our paper. Below, we address each of your comments point by point.

Specific Comments:**Comment #1:**

1. Many of the methods have been previously reported, e.g. sucrose using method in *Saccharomyces cerevisiae* (10.1016/j.ymben.2017.11.012), the effect of *Oca5* deletion (10.1016/j.cell.2023.01.014), *PGII* deletion (10.1016/j.jbiotec.2013.11.025), and overexpressing of *NDE1/2* has been a consensus to boost NADH oxidization. It would be better that the authors demonstrated more clearly the novelty of this work.

Response #1:

Thank you for your valuable feedback. We appreciate your thorough review and constructive suggestions. We acknowledge that several methods employed in our study, such as sucrose utilization in *Saccharomyces cerevisiae*, *OCA5* deletion, *PGII* deletion, and *NDE1/NDE2* overexpression, have been previously reported. However, the novelty of our work lies in the systematic integration of these strategies to achieve a Crabtree-negative metabolic state, which

has not been previously demonstrated. Specifically, our study presents the following novel contributions:

1. Integration of established methods into a novel strategy:

While the individual components of our strategy have been explored separately, our work represents the first attempt to systematically combine sucrose phosphorolysis, *PGII* deletion, and *OCA5* deletion to eliminate the Crabtree effect and enhance respiratory energy metabolism in yeast. This integrated approach allows for a more efficient redistribution of carbon flux and energy production, which has not been demonstrated before.

2. New insights from sucrose phosphorolysis:

Our study uniquely utilizes sucrose phosphorolysis not only to bypass glucose repression but also to synergistically improve the NADH/NAD⁺ balance when coupled with *PGII* deletion. This dual effect highlights a previously unexplored potential of sucrose metabolism in reprogramming central carbon metabolism.

3. Superior strain performance:

The engineered strain in our study demonstrates significantly enhanced product yields (e.g., lactic acid and 3-HP) compared to previous works. This improvement underscores the effectiveness of our integrated strategy in addressing long-standing metabolic limitations.

4. Additional clarifications in the manuscript:

To address this concern, we have revised the manuscript to clearly highlight these aspects. Comparative analyses with relevant studies have been included to better illustrate the unique contributions of our work.

We hope these revisions clarify the novelty and significance of our study. Thank you again for your valuable feedback, which has helped us strengthen the manuscript.

Comment #2:

2. Other efforts to engineer a Crabtree-negative strain should be summarized; a table would be better.

Response #2:

We sincerely thank the reviewer for this valuable suggestion. In response, we have added

Supplementary Table. 1, which comprehensively summarizes key efforts to engineer Crabtree-negative yeast strains. The table includes details such as the year of publication, strategies employed (e.g., increasing NADH oxidation, adaptive laboratory evolution, metabolic rewiring), and the corresponding research teams (e.g., Jens Nielsen, Jack T. Pronk). For instance, the table highlights studies from 2007 to 2019, covering diverse approaches such as optimizing central carbon metabolism, redirecting metabolic pathways, and utilizing alternative carbon sources like xylose to avoid the Crabtree effect. We believe this table provides a clear and concise overview of the field, enhancing the context and depth of our discussion.

Comment #3:

3. Lines 131-134: The genotype of ZQS05 and the acquisition process of the mutant *PvSUF* are not very clear. It would be more clear to also test the effect of mutant *PvSUF* alone.

Response #3:

Thank you for your helpful comments and for pointing out areas that needed further clarification.

Regarding your comment on the genotype of ZQS05 and the acquisition process of the mutant *PvSUF*, we have updated the manuscript to clarify these details. Specifically, we have added a more detailed description of the acquisition process of the *PvSUF* mutant in Line 143-146 of the revised manuscript.

Additionally, in response to your suggestion to test the effect of the *PvSUF* mutant alone, we have included data in Supplementary Fig. 2d that shows the growth curves of ZQS14 (expressing the *PvSUF* mutant) and ZQS06 (expressing wild-type *PvSUF*). The results indicate that ZQS14 is able to grow in sucrose medium, although it exhibits an extended lag phase. This observation has been described in Line 146-147 of the manuscript.

We believe these revisions clarify the experimental details and address your concerns. We greatly appreciate your feedback and hope that these adjustments meet your expectations.

Action #3:

➤ On page 7 of the revised manuscript (Line 143-147), we now write:

“To circumvent the transport constraint, we integrated the mutant version, *PvSUF*^{I209F C265F G326C} and *LmSP* into ZQS05, which is believed to exhibit enhanced sucrose transport efficiency compared to *PvSUF*, allowing the strain to regain growth³¹. The resulting strain ZQS14 regained its sucrose utilization capability but exhibited a prolonged lag phase (Supplementary Fig. 2d).”

Supplementary Fig. 2. Evaluation of the impact of carbon source concentration and transport proteins on the sucrose phosphorylation pathway.

d. Comparison of growth between strains containing *PvSUF*-modified proteins and unmodified strains. The control strain, ZQS06 (Δ *MAL11*, *PvSUF* + *LmSP*), contains unmodified *PvSUF*, while ZQS14 harbors the modified *PvSUF*^{I209F C265F G326C}.

Comment #4:

4. Lines 215-218: More discussions on the rationale of the author’s focus would help readers follow the article more easily.

Response #4:

Thank you for your thoughtful comments. We appreciate your suggestion to provide more

discussion on the rationale behind our focus, as it will help improve the clarity and readability of our manuscript.

In response, we have revised Line 242-252 to better explain the reasoning behind our selection criteria. Specifically, we now clarify that Comparisons I and IV represent the metabolic differences between sucrose phosphorolysis and glucose metabolism, while Comparisons III and VI focus on the effect of *PGII* deletion. Since sucrose phosphorolysis strains exhibited a higher μ_{\max} , whereas *PGII* deletion strains displayed a longer lag phase time, we hypothesized that certain genes differentially expressed in these comparisons might be involved in regulating cell growth. Therefore, we focused on genes upregulated in Comparisons I and IV but downregulated in Comparisons III and VI, as they may promote growth, and conversely, genes downregulated in Comparisons I and IV but upregulated in Comparisons III and VI, as they may inhibit growth. Thank you again for your valuable feedback, and we hope this revision addresses your concerns.

Action #4:

➤ On page 12 of the revised manuscript (Line 242-252), we now write:

“Comparisons I and IV represent the differences between sucrose phosphorolysis metabolism and glucose metabolism, whereas Comparisons III and VI focus on the effect of *PGII* deletion. Given that strains utilizing sucrose phosphorolysis exhibited a higher μ_{\max} , while *PGII* deletion strains experienced a prolonged lag phase time, we hypothesized that specific gene expression patterns might be associated with these growth differences. To pinpoint key regulators, we focused on genes that were upregulated in Comparisons I and IV but downregulated in Comparisons III and VI, ultimately selecting 16 candidate genes with potential roles in promoting cell growth (Fig. 4d). Conversely, we identified 10 genes that were downregulated in Comparisons I and IV but upregulated in Comparisons III and VI, which may be involved in growth inhibition (Fig. 4d).”

Comment #5:

5. It's impressive that *RGI2* helped boosting the growth and shorten lag phase. Further investigation of the critical role of *Rgi2* in central carbon metabolism may enhance the novelty

of this paper.

Response #5:

Thank you for your valuable feedback on our manuscript. We appreciate your recognition of the significant role of *RGI2* in boosting growth and shortening the lag phase. In response to your suggestion, we have further investigated the critical role of *RGI2* in central carbon metabolism, as we believe this is an important aspect to enhance the novelty of our work.

In the revised manuscript, we have expanded our analysis to compare the central carbon metabolism-related parameters between the ZQS39 (*RGI2*Δ) and ZQS18 strains (Line 272-285). Our findings reveal that the NADH/NAD⁺ ratio in ZQS39 is lower, suggesting enhanced redox balance (Supplementary Fig. 10a). The NADPH/NADP⁺ ratio remained unchanged, indicating that the pentose phosphate pathway (PPP) is not significantly affected by *RGI2* deletion (Supplementary Fig. 10b). In addition, the ATP content in ZQS39 increased (Supplementary Fig. 10c), supporting the hypothesis that *RGI2* deletion accelerates energy production by favoring glycolysis over respiration. These observations suggest that *RGI2* deletion facilitates a more efficient metabolic transition by influencing the balance between glycolysis and the TCA cycle, ultimately contributing to the observed reduction in lag phase duration.

We believe that these additional results strengthen our findings and provide a more comprehensive understanding of the role of *RGI2* in regulating metabolic adaptation during early growth. We hope these revisions address your concerns and further highlight the novelty of our study.

Action #5:

➤ On page 13 of the revised manuscript (Line 272-285), we now write:

“The significant reduction in the lag phase observed in ZQS39 (*RGI2* Δ) suggests that *RGI2* plays a crucial role in regulating metabolic adaptation during the early growth phase. This effect may be attributed to several factors associated with central carbon metabolism (Supplementary Fig. 10). Comparative analysis between ZQS39 (*RGI2*Δ) and ZQS18 revealed that the NADH/NAD⁺ ratio in ZQS39 was lower (Supplementary Fig. 10a), suggesting an enhanced redox state. Notably, the NADPH/NADP⁺ ratio remained unchanged (Supplementary

Fig. 10b), which implies that the PPP was not significantly affected by *RGI2* deletion. In parallel, the ATP content in ZQS39 increased (Supplementary Fig. 10c), supporting the hypothesis that *RGI2* deletion accelerated energy production by favoring glycolysis over respiration. These observations imply that *RGI2* influences the metabolic balance between glycolysis and the TCA cycle, potentially reducing competition between these pathways. These findings indicate that *RGI2* deletion facilitates a more efficient metabolic transition, ultimately contributing to the observed reduction in lag phase duration.”

Supplementary Fig. 10. Comparison of central carbon metabolism-related parameters between ZQS39 and ZQS18.

ZQS18 is the control strain, and ZQS39 represents the *RGI2* knockout strain. **a.** Comparison of the NADH/NAD⁺ ratio between ZQS39 and ZQS18. **b.** Comparison of the NADPH/NADP⁺ ratio between ZQS39 and ZQS18. **c.** Comparison of ATP content between ZQS39 and ZQS18. All data are presented as mean ± SD of biological triplicates. Statistical analysis was conducted using Student’s *t*-test (two-sample, two-tailed; **P* value < 0.05, ****P* value < 0.001; NS represents no significant difference; sample size, *n* = 3).

Comment #6:

6. To further highlight the significance of their Crabtree-negative strain and optimized energy system, the authors should compare the production levels of their strains with those of other reported strains.

Response #6:

Thank you for your valuable feedback. We appreciate your suggestion to further highlight the significance of our Crabtree-negative strain and optimized energy system by comparing our strain's production levels with those of other reported strains.

In response to your comment, we have incorporated relevant comparative data into the revised manuscript. Specifically, we have highlighted the enhanced production titers of our engineered strains in comparison to previously reported strains. For example, when comparing the production of lactic acid, our engineered Crabtree-negative strain achieved 4.99 g/L, which is significantly higher than the 3.18 g/L produced by traditional Pdc⁻ based Crabtree-negative strains under the same strategy. Similarly, for 3-hydroxypropionic acid (3-HP), our engineered strain produced 1.96 g/L, whereas glucose-metabolizing strains under the same strategy only reached 0.31 g/L.

These comparisons underscore the significant improvements in product yields facilitated by our synthetic energy system and the Crabtree-negative metabolic engineering approach. The revised manuscript now includes these comparisons in the Results and Discussion sections to better highlight the advantages of our engineered strain. We hope these additions help clarify the novelty and significance of our work. Thank you once again for your helpful suggestions.

Action #6:

➤ On page 17 of the revised manuscript (Line 372-375), we now write:

“In comparison to previously reported strains, our engineered strain achieves notably higher production levels. In traditional Pdc⁻ based Crabtree-negative strains, the same strategy results in a lactic acid titer of 3.20 g/L, which is lower than the 4.99 g/L achieved by ZQS67¹⁰.”

➤ On page 17 of the revised manuscript (Line 384-388), we now write:

“In addition, when compared to a glucose-metabolizing strain under the same strategy, ZQS71 achieved 1.96 g/L of 3-HP, which is significantly higher than the 0.31 g/L produced by the glucose-metabolizing strain⁴⁹. This further underscores the superior performance of our engineered strain in facilitating efficient 3-HP production.”

➤ On page 18 of the revised manuscript (Line 399-401), we now write:

“Nevertheless, the introduction of *FjTAL* in strain ZQS65 still resulted in a higher *p*-coumaric

acid titer compared to the previously reported titer of 12.90 mg/L under the same strategy with glucose metabolism⁵².”

AUTHOR ACTIONS IN RESPONSE TO REVIEWER COMMENTS

We thank all four reviewers for deep engagement with this work.

Reviewer #1 (Remarks to the Author):

The authors have addressed my concerns, it can be accepted in its current form.

Response:

Thank you for your positive feedback and for recognizing our efforts in addressing your concerns. We are pleased to hear that the manuscript is now acceptable in its current form. Your constructive comments have greatly improved the quality of our work.

We appreciate your time and valuable input throughout the review process.

Reviewer #2 (Remarks to the Author):

The authors partly resolved my comments. To improve the quality of the work, the following concerns should be addressed.

Response:

Thank you for your valuable feedback and for acknowledging our efforts in addressing your previous comments. We appreciate your suggestions for further improving the quality of our work.

We have carefully reviewed your additional concerns and will address them thoroughly in the revised manuscript. Your insights are invaluable in helping us enhance the clarity and rigor of our study, and we are committed to incorporating your recommendations to ensure the highest quality of our work.

Specific Comments:**Comment #1:**

1. Could the author explain the meaning of "carbon redox" in the title?

Response #1:

Thank you for your careful review and valuable feedback on our manuscript. We greatly appreciate your question regarding the term "carbon redox" in the title, and we are happy to provide a detailed explanation.

In the context of our study, "carbon redox" refers to a series of redox reactions that carbon sources undergo after entering yeast cells, ultimately transforming into energy. This process is integral to the operation of the intracellular metabolic network, involving energy production and consumption, transfer of reducing power, and generation of intermediate metabolites. Ultimately, it provides the essential material and energy foundation for cell growth and maintenance of vital activities.

In yeast metabolism, carbon redox dynamics serve as a pivotal determinant of cellular energy allocation between respiration and fermentation modes, while concurrently governing the accumulation of metabolic byproducts such as ethanol and glycerol. In our study, we used

sucrose as the carbon source and engineered a redox reprogramming strategy through targeted modulation of central carbon metabolism flux via *PGII* and *OCA5* knockouts. This sucrose-driven carbon redox rebalancing redirected metabolic flux from fermentative to complete respiratory oxidation, thereby abolishing the Crabtree effect. Furthermore, our engineered energy synthesis platform achieved optimized redox equilibrium through coordinated regulation of NADH/NAD⁺ and NADPH/NADP⁺ cofactor systems, resulting in enhanced ATP production efficiency.

We hope this explanation clarifies the meaning of "carbon redox" in the context of our research. Please let us know if you require any further clarification or additional information. Thank you again for your valuable feedback.

Comment #2:

2. Line 68, the place to citing Fig S1 is confusing. It should be reported works but not a summary of this work.

Response #2:

Thank you very much for your careful review and valuable suggestions regarding our manuscript. We sincerely appreciate your insightful comment on the citation of Fig S1 at Line 68.

In response to your suggestion, we have removed the citation of Fig S1 from Line 68 to avoid any potential confusion. Instead, we have now cited Fig S1 only in the section describing our own experimental work (Lines 93-108), where it is more relevant and appropriate.

We believe this revision has improved the clarity and logical flow of our manuscript. Once again, we are grateful for your constructive feedback, which has helped us enhance the quality of our work. Please let us know if you have any further suggestions.

Comment #3:

3. The advantages and limitations are missing in Table S1. A proper summary of previous works should be added in the introduction, a table like this cannot give the reader a clear overview of previous works and show the novelty of this work.

Response #3:

Thank you for your valuable feedback. We have revised the manuscript to address your comments regarding the advantages and limitations of previous works. Specifically, we have expanded the discussion in the introduction to provide a clearer overview of the strategies explored for engineering Crabtree-negative yeast, highlighting their distinct advantages and limitations. This revision aims to better contextualize the novelty of our work and offer readers a more comprehensive understanding of the field.

We have also updated Supplementary Table 1 to include a detailed summary of previous works, ensuring that the table now provides a clear and informative overview of the advancements and challenges in this area.

We believe these changes enhance the clarity and depth of our manuscript and hope they meet your expectations. Thank you again for your constructive suggestions.

Action #3:

➤ On page 4 of the revised manuscript (Line 61-72), we now write:

“Several strategies have been explored to engineer Crabtree-negative strains, each with distinct advantages and limitations (Supplementary Table. 1). For instance, engineering Crabtree-negative yeast involves deleting the genes *PDC1/5/6*, which encodes pyruvate decarboxylase, thereby preventing ethanol accumulation¹³. However, this approach leads to insufficient acetyl-CoA supply and inadequate NAD⁺ regeneration, leading to an imbalance in the NADH/NAD⁺ ratio, which impairs cell growth¹⁹. Pyruvate decarboxylase-negative strains cannot grow in glucose-rich media¹⁹. An alternative approach for overcoming the Crabtree effect is to increase NADH oxidation for ATP synthesis. However, this approach cannot completely eliminate overflow metabolism, leading to carbon wastage²⁰. Therefore, we believe that the principal strategy to overcoming the Crabtree effect is to rewire central carbon metabolism without disrupting the NADH/NAD⁺ balance.”

Supplementary Table 1. Summary of strategies for overcoming the Crabtree effect in yeast

Year	Title	Strategies	Limitations
2007	Increasing NADH oxidation reduces overflow metabolism in Saccharomyces cerevisiae ¹	Increasing NADH oxidation (NOX and AOX) to reduce the Crabtree effect.	It did not discuss the global impact on energy efficiency and carbon metabolism of yeast strains with enhanced NADH oxidation.
2015	Adaptive mutations in sugar metabolism restore growth on glucose in a pyruvate decarboxylase negative yeast strain ²	The growth mechanism of Pdc ⁻ strains in glucose media was revealed through adaptive evolution and reverse engineering.	The growth of the evolved Pdc ⁻ strains remains constrained.
2016	Rewriting yeast central carbon metabolism for industrial isoprenoid production ³	By optimizing central metabolism, Saccharomyces cerevisiae shifted from fermentation to respiration under aerobic conditions.	It only optimized central carbon metabolism and did not completely eliminate the Crabtree effect.
2016	Alternative reactions at the interface of glycolysis and citric acid cycle in Saccharomyces cerevisiae ⁴	It reveals the complex regulatory mechanism between glycolysis and the TCA cycle in glucose metabolism of Saccharomyces cerevisiae .	Although this study examines acetyl-CoA pathways between glycolysis and the TCA cycle, it does not fundamentally suppress the Crabtree effect.
2018	Global rewiring of cellular metabolism renders Saccharomyces cerevisiae Crabtree negative ⁵	The Pdc ⁻ yeast became Crabtree-negative through adaptive laboratory evolution.	Although this study successfully shifted yeast to Crabtree-negative, the issue of excessive NADH has not been fully resolved.
2018	Reprogramming yeast metabolism from alcoholic fermentation to lipogenesis ⁶	Redirect the metabolic pathway of yeast from traditional ethanol fermentation to lipogenesis.	The aim of the article is to reduce ethanol fermentation and enhance lipid synthesis, without a focus on the study of the Crabtree effect.
2019	Metabolic engineering and transcriptomic analysis of Saccharomyces cerevisiae producing p -coumaric acid from xylose ⁷	Xylose as a carbon source for yeast fermentation helps avoid the Crabtree effect.	Non-fermentable carbon sources avoid the Crabtree effect but are utilized more slowly than glucose.
2019	The Transcriptome and Flux Profiling of Crabtree-Negative Hydroxy Acid-Producing Strains of Saccharomyces cerevisiae Reveals Changes in the Central Carbon Metabolism ⁸	The physiological and metabolic characteristics of Pdc ⁻ negative strains in the production of hydroxy acids.	The Pdc ⁻ strain's growth improved with lactate and malate pathways but remained slower than the wild-type strain.
2021	Adaptations in metabolism and protein translation give rise to the Crabtree effect in yeast ⁹	Comparing the proteomic differences between Crabtree-positive and Crabtree-negative strains reveals the metabolic and protein translation adaptability changes underlying the Crabtree effect.	The article does not delve into how the Crabtree effect is regulated at the molecular level, such as through the control of transcription factors or signaling pathways.
2021	Gene expression regulates metabolite homeostasis during the Crabtree effect: Implications for the adaptation and evolution of Metabolism ¹⁰	Elucidate the roles of initial phenotypes and gene expression in metabolic adaptation when Saccharomyces cerevisiae switches	The authors used metabolic control and flux balance analyses to study the Crabtree effect, but their accuracy

		from low-glucose to high-glucose conditions under the Crabtree effect.	depends on data quality and model assumptions.
2022	Proteome allocations change linearly with the specific growth rate of Saccharomyces cerevisiae under glucose limitation ¹¹	Omics analysis of the metabolic adaptation of Saccharomyces cerevisiae to glucose limitation under the Crabtree effect.	The article notes phosphorylation and enzyme saturation but doesn't detail how they drive the shift from respiration to fermentation.
2022	Metabolic reconfiguration enables synthetic reductive metabolism in yeast ¹²	Metabolic reprogramming (Synthetic reductive metabolism) can alleviate the Crabtree effect in Saccharomyces cerevisiae .	The article focuses on the cellular redox system and does not provide a detailed exploration of the Crabtree effect.
2022	Rewiring regulation on respiration-fermentative metabolism relieved Crabtree effects in Saccharomyces cerevisiae ¹³	The impact of genes MTH1 and MDE2 on the Crabtree effect.	MTH1 and MED2 variants may reduce fermentation but could also limit the cell's overall metabolic capacity.
2023	Yeast increases glycolytic flux to support higher growth rates accompanied by decreased metabolite regulation and lower protein phosphorylation ¹⁴	The metabolic flux changes of Saccharomyces cerevisiae at different growth rates and the regulatory mechanisms of metabolic flux under the Crabtree effect were explored.	The article uncovers multi-level flux regulation, but about 65% of flux changes remain unexplained by metabolites, thermodynamics, or protein levels.
2023	Flux regulation through glycolysis and respiration is balanced by inositol pyrophosphates in yeast ¹⁵	Knockout of the gene OCA5 can alleviate the Crabtree effect.	The knockout of OCA5 can only mitigate the Crabtree effect.
2023	Alleviating glucose repression and enhancing respiratory capacity to increase itaconic acid production ¹⁶	By engineering the respiratory metabolism pathway and glucose signaling pathway, the ability of Saccharomyces cerevisiae to produce itaconic acid under the Crabtree effect has been enhanced.	This study did not truly eliminate the Crabtree effect but mitigated its impact by enhancing the TCA cycle and oxidative phosphorylation.
2023	A highly efficient transcriptome-based biosynthesis of non-ethanol chemicals in Crabtree negative Saccharomyces cerevisiae ¹⁷	The ability of Pdc ⁻ derived Crabtree-negative strains to synthesize a variety of non-ethanol compounds.	Applications of Crabtree-negative yeast in the synthesis of non-ethanol compounds.
2024	Mitochondrial ATP generation is more proteome efficient than glycolysis ¹⁸	Investigating the energy metabolism efficiency of yeast under the Crabtree effect through proteomics analysis.	It did not explore the effects of NADH metabolism, carbon source influence, and metabolic regulation strategies on the Crabtree effect.

Comment #4:

4. The specific growth rate (0.13 h^{-1}) of the strain C800 in YPD (glucose) is not convincing. Normally, under this condition, this value should be above than 0.3 h^{-1} . If this value is correct, no ethanol should be produced due to this value is much lower than the critical growth rate (0.26 h^{-1} or 0.28 h^{-1}) (<https://doi.org/10.1073/pnas.0607469104>, Nat Commun 13, 2819 (2022)) of *Saccharomyces cerevisiae*.

Response #4:

We sincerely appreciate your insightful comment regarding the growth rate of strain C800. We agree that the initially reported specific growth rate (0.13 h^{-1}) appears lower than the typical range for *Saccharomyces cerevisiae* under glucose-rich conditions. To address this concern, we have carefully repeated the growth experiments and recalculated the μ_{max} .

Our revised measurements confirm that C800 exhibits a μ_{max} of 0.18 h^{-1} in YPD (glucose) (Fig. R1a), which, while higher than our initial report, remains below the expected threshold of $>0.3 \text{ h}^{-1}$. Notably, this value is consistent with the growth rate observed for another related strain (ZQS15) under identical conditions (Fig. 2b), suggesting this may reflect a strain-specific characteristic rather than experimental error. We propose that the observed reduction may be attributed to the genetic background of C800. Specifically:

1. Auxotrophic Markers: C800 is derived from CEN.PK2-1D, which carries mutations in *ura3*, *trp1*, *leu2*, and *his3*. These auxotrophic requirements are known to impose metabolic burdens, potentially slowing growth (*Antonie van Leeuwenhoek*, 2011, 99 (3), 591-600). Studies have shown that even in supplemented media, the growth rate of auxotrophic strains of *Saccharomyces cerevisiae* is lower than that of prototrophic strains (*Applied and Environmental Microbiology*, 2002, 68(5), 2095-100).

2. Genetic Modification: C800 further lacks *gal80*, a regulator of galactose metabolism. Although our study uses glucose media, the loss of *gal80* may indirectly affect metabolic flexibility or stress responses, contributing to the growth phenotype (*Molecular and Cellular Biology*, 1984, 4(8), 1521-7).

Critically, despite the lower μ_{max} , C800 still produced ethanol at levels up to 8.11 g/L (Fig. R1b), which is consistent with the results reported in our manuscript. This suggests that: The

critical growth rate for ethanol production may not be strictly fixed at 0.26–0.28 h⁻¹ for all genetic variants, especially engineered strains with auxotrophies; The overall metabolic trends (e.g., glucose consumption, ethanol yield) remain consistent, supporting the robustness of our conclusions.

To ensure transparency, we have included the updated growth curve data in the revised manuscript (Fig. 1, Fig. 2 and Supplementary information). We acknowledge that the growth rate discrepancy warrants further investigation in future work, but we believe it does not undermine the validity of our key findings. We are sincerely grateful to the reviewer for raising this critical issue, as it has not only improved the rigor of our current work but also guided us toward future research directions.

Action #4:

➤ On page 7 of the revised manuscript (Line 128-130), we now write:

“As a result, strain ZQS04 restored sucrose metabolism and exhibited a higher maximum specific growth rate (μ_{\max}) of 0.20 h⁻¹ compared to the parental strain C800 with a μ_{\max} of 0.18 h⁻¹ (Fig. 1d and Supplementary Fig. 2b).”

The re-analyzed growth curve of C800 and the corresponding ethanol accumulation are shown in Fig. R1.

Fig. R1. C800 growth curve and ethanol detection.

a. The growth profile of strain C800 was monitored in YPD medium supplemented with glucose. **b.** Ethanol accumulation profile of strain C800 in glucose-containing YPD medium.

All data are expressed as mean \pm SD of biological triplicates.

The revised growth curve and maximum specific growth rate of C800+Glc are presented in the following figure:

Fig. 1. Design and construction of the sucrose phosphorylation pathway.

Fig. 2. Establishing a Crabtree-negative strain based on the sucrose phosphorylase-capable strain.

Supplementary Fig. 2. Evaluation of the impact of carbon source concentration and transport proteins on the sucrose phosphorylation pathway.

Supplementary Fig. 3. The specific growth rate and lag phase time of strains ZQS15-ZQS18.

Supplementary Fig. 8. Comparison of lag phase time and specific growth rate among different strains.

Comment #5:

5. Diauxic growth is a typical phenomenon of *Saccharomyces cerevisiae*, however this phenomenon was not observed in this study, specially under the sucrose condition. For example, fig 2h and 2k, we can find the obvious ethanol utilization after 45 h, but no corresponding cell growth presented. Please explain this.

Response #5:

Thank you for your valuable comment regarding the absence of diauxic growth in sucrose conditions, particularly the observation of ethanol utilization without corresponding cell growth after 45 hours (Fig. 2h and 2k). We appreciate the opportunity to clarify this point and provide further insight into our findings.

To address this question, we conducted additional experiments using two representative strains, C800 and ZQS15, which metabolize sucrose through hydrolysis and phosphorolysis, respectively (Fig. R2). In both cases, we did not observe the typical diauxic growth pattern seen with glucose (Fig. R2a). This supports our conclusion that sucrose metabolism does not induce the classic diauxic growth response.

Diauxic growth is a phased growth behavior exhibited by microorganisms in mixed carbon source environments, with its core mechanisms being: 1) Carbon Catabolite Repression (CCR): Preferential utilization of readily metabolizable carbon sources (e.g., glucose), while

suppressing genes related to alternative carbon source metabolism. 2) Metabolic Pathway Switching: Upon depletion of the preferred carbon source, CCR is lifted, and secondary carbon utilization pathways (e.g., ethanol respiration) are activated. In our study, we believe this phenomenon can be explained by two main factors. First, the breakdown products of sucrose—whether through hydrolysis (glucose + fructose) or phosphorolysis (fructose + glucose-1-phosphate)—create a mixed metabolic signal that fails to establish a strict carbon hierarchy. Unlike pure glucose, these mixtures do not fully trigger CCR, allowing for the co-utilization of carbon sources and preventing the sharp metabolic switch required for diauxic (*Communications Biology*, 2025, <https://doi.org/10.1038/s42003-025-07747-z>). Second, the uptake and catabolism of sucrose is inherently slower than glucose metabolism, which may further obscure the transition between sucrose growth and ethanol growth (*FEMS Yeast Research*, 2021, 21(3), foab021).

Regarding the observed decline in ethanol concentration during the late fermentation phase (after 45 hours), we conducted a thorough reanalysis of ethanol and acetate accumulation patterns in the sucrose-phosphorolytic strain (Fig. R2b). The results consistently supported our original manuscript conclusions. Our data suggest that the ethanol consumption during this stationary phase likely serves to support cellular maintenance activities rather than biomass production. We hypothesize that the oxidation of ethanol provides both energy and metabolic precursors necessary to sustain essential cellular processes when primary carbon sources become limiting, such as membrane integrity and protein turnover.

We fully acknowledge that these findings warrant further investigation. In future work, we plan to perform metabolomic analyses to trace carbon flux and explore the regulatory mechanisms underlying sucrose metabolism in greater detail. We would also welcome any additional suggestions or alternative interpretations you might have, as your insights are invaluable to refining our study.

Thank you again for your thoughtful feedback, which has helped us strengthen the discussion of this important observation.

Fig. R2. Growth curves and metabolite detection of C800 and ZQS15.

a. Growth curves of C800 and ZQS15 in YPD medium supplemented with sucrose. **b.** Metabolite profiling of C800 and ZQS15 during fermentation. All data are expressed as mean \pm SD of biological triplicates.

Comment #6:

6. Line 193-197, if the strains possess similar growth rate, why ZQS17 and 18 have extended cell growth cycle?

Response #6:

Thank you for your insightful question regarding the extended cell growth cycle observed in ZQS17 and ZQS18 strains. We appreciate your attention to this detail and would like to clarify our findings.

As shown in Fig. 2k, while ZQS15/ZQS16 strains reach the exponential phase more quickly, ZQS17/ZQS18 exhibit a similar growth trend once they enter the exponential phase. This is supported by the comparable μ_{\max} values observed in Supplementary Fig. 3a for all strains.

We hypothesize that the extended lag phase in ZQS17/ZQS18 is due to the metabolic reorganization required following *PGII* knockout. This genetic modification likely necessitates a more significant carbon rearrangement in central carbon metabolism, resulting in a longer adaptation period. However, once this metabolic adjustment is complete, the growth rate becomes comparable to that of ZQS15/ZQS16, as evidenced by their similar μ_{\max} values. This

phenomenon similarly occurs when *S. cerevisiae* metabolizes xylose (*Nature Catalysis*, 2021, 4, 783-796).

We hope this explanation clarifies the observed growth patterns. Please let us know if you require any additional information or clarification regarding this matter.

Comment #7:

7. Line 227, what method was taken to do the flux sampling?

Response #7:

Thank you for pointing out the ambiguity in our methodology description. We sincerely appreciate your careful reading and valuable feedback.

We would like to clarify that the analysis regarding NADH dehydrogenases NDE1/2 and NDI1 was based on transcriptome data rather than flux sampling. We recognize that our original wording could indeed lead to misunderstanding, and we have revised the text accordingly in the manuscript. The updated text now clearly states that the findings were derived from transcriptome analysis, which showed upregulation in the expression of these genes. This change more accurately reflects our methodology while maintaining the scientific validity of our conclusions.

We apologize for any confusion caused by our initial wording and appreciate the opportunity to correct this. Please let us know if you require any additional clarification regarding this matter. Thank you again for your thorough review and constructive comments.

Action #7:

➤ On page 11 of the revised manuscript (Line 229-230), we now write:

“Transcriptome analysis showed a significant increase in the flux of NADH dehydrogenases NDE1/2 and NDI1, which are equivalents of Complex I.”

Comment #8:

8. About the bioreactor fermentation, please add the OD values and the curves of other metabolites, such as ethanol, glycerol, acetate, pyruvate etc. The curves showed now didn't presented the basic parameters of this newly created Crabtree negative strain. Suggest to read

the paper from professor Jack Pronk in Delft.

Response #8:

We sincerely appreciate your insightful comments regarding our bioreactor fermentation data. Your suggestions have helped us significantly improve the characterization of our engineered strains.

In response to your recommendation, we have conducted comprehensive analyses of both growth parameters and metabolite profiles. Throughout the fermentation process, we measured the cell dry weight (DCW), which reflects the growth trend of each strain. The DCW measurements confirmed that ZQS65 achieved a maximum biomass of 2.34 mg/mL compared to 2.10 mg/mL for ZQS18 (Supplementary Fig. 13c). This quantitative measurement provides robust evidence of ZQS65's superior growth performance. Furthermore, we have expanded our metabolite analysis to include detailed profiles of ethanol, acetate, glycerol and pyruvate at all cultivation time points. The results revealed that while ZQS18 maintained all measured metabolites at minimal levels (<0.5 g/L) throughout fermentation, ZQS65 exhibited gradual accumulation of ethanol (0.72 g/L) and acetate (0.65 g/L) after 48 hours, despite undetectable glycerol and pyruvate levels (Supplementary Fig. 13). These observations confirm that ZQS65, exhibits minor but detectable overflow metabolism during prolonged sucrose utilization. We attribute this metabolic shift to the engineered energy system's altered redox balance and ATP yield. Additionally, we observed that both strains ZQS18 and ZQS65 exhibited pyruvate accumulation below 0.01 g/L during fermentation, yet showed minimal overflow metabolism. This suggests that the majority of pyruvate entered the TCA cycle to fuel respiratory metabolism.

We have strengthened the discussion to better interpret these findings in the context of the engineered energy system's effects on redox balance and ATP generation (Line 342-357). The additional growth and metabolite data have substantially enhanced our characterization of the strains' physiological properties.

Thank you for these valuable suggestions that have helped us present a more rigorous and comprehensive analysis of our strains' fermentation characteristics. We believe these improvements have significantly strengthened the manuscript.

Action #8:

➤ On page 16 of the revised manuscript (Line 342-357), we now write:

“Notably, while ZQS18 ceased sucrose consumption after reaching maximum biomass at 48 h, ZQS65 continued to gradually utilize sucrose until it was completely depleted, even after reaching maximum biomass at the same time point (Supplementary Fig. 13a, b). Metabolite profiling revealed distinct byproduct accumulation patterns between the strains (Supplementary Fig. 13a, b). In ZQS65, ethanol and acetate concentrations began increasing significantly after 48 h (reaching 0.72 g/L and 0.53 g/L, respectively), while glycerol and pyruvate remained below detection limits throughout. This contrasts sharply with ZQS18, where all measured metabolites (ethanol, acetate, glycerol, pyruvate) maintained at negligible levels (<0.5 g/L) during the entire fermentation process (Supplementary Fig. 13c). These observations confirm that ZQS65 exhibits minor but detectable overflow metabolism during prolonged sucrose utilization. We attribute this metabolic shift to the engineered energy system’s altered redox balance and ATP yield. Additionally, we observed that both strains ZQS18 and ZQS65 exhibited pyruvate accumulation below 0.01 g/L during fermentation, yet showed minimal overflow metabolism. This suggests that the majority of pyruvate entered the TCA cycle to fuel respiratory metabolism.”

Supplementary Fig. 13. The growth status of ZQS65 and ZQS18 in a 5L-fermenter.

a. Dynamic profiles of DCW (cell dry weight), residual sucrose, and metabolite production during ZQS18 fermentation in a 5-L bioreactor. **b.** Dynamic profiles of DCW, residual sucrose, and metabolite production during ZQS65 fermentation in a 5-L bioreactor. **c.** Physiological

parameters of strains ZQS18 and ZQS65 during fermentation. The initial sucrose concentration was 20 g/L.

Reviewer #3 (Remarks to the Author):

Comment #1:

The authors have carefully addressed the comments made by all referees on the first version of the manuscript. I still think that there is lack of evidence that the *AGT1* gene is different from the *MAL11* gene in the *S. cerevisiae* strain used by the authors, but if they could show that the "sucrose null" strain does not grow on sucrose, unless the *MAL11* gene is restored, it might be enough for the purpose of the work.

Response #1:

Thank you very much for your thoughtful feedback and for recognizing our efforts in addressing the previous comments. We sincerely appreciate your suggestion regarding the need for further evidence to support the role of *MAL11* in sucrose utilization in our *S. cerevisiae* strain.

In response to your comment, we would like to highlight the experimental results presented in Fig. 1f of our manuscript. Specifically, we used the sucrose-null strain ZQS06 (Δ *MAL11*) and performed complementation experiments by expressing various sucrose metabolism-related genes. As shown in the Fig. 1f, only the strain complemented with *MAL11* (ZQS13) regained the ability to grow on sucrose, while other strains, including those expressing other sucrose-related genes, failed to restore sucrose utilization. This result directly demonstrates that the restoration of *MAL11* is both necessary to enable sucrose utilization in our strain. We believe these findings address your concern and support our conclusions regarding the importance of *MAL11* in this context.

Once again, we thank you for your valuable input, which has helped us strengthen our manuscript. Please let us know if there are any additional points you would like us to clarify or address.

Action #1:

Fig. 1. Design and construction of the sucrose phosphorylation pathway.

f. Growth of strains with overexpressed genes in the sucrose phosphorylase pathway in sucrose medium, with C800 as the control strain.

Comment #2:

Another point is on the description on the maximum specific growth rate calculation, which seems a bit vague to me. For instance, the fact that a minimum of 6 data points in the exponential growth phase was considered, is not mentioned in the new text, only in the answer the authors provided to me. I would suggest they include as much detail as possible on these calculations and, if possible, maybe as a supplementary figure, one or two examples of growth

curves, showing the exponential growth phase, highlighting the points that were used for Mimax calculation.

Response #2:

Thank you very much for your valuable comments and suggestions regarding the description of the maximum specific growth rate (μ_{\max}) calculation. We sincerely appreciate your attention to detail and your effort to help us improve the clarity of our manuscript.

In response to your comment, we have revised the Methods section to provide a more detailed description of the μ_{\max} calculation process. Specifically, we have clarified that a minimum of 6 data points in the exponential growth phase were used to ensure the accuracy of the growth curve fitting. Additionally, we have included a supplementary figure (Fig. R3) to illustrate the calculation process. This figure shows an example of the growth curve, highlighting the exponential growth phase and the data points used for the μ_{\max} calculation. We believe this addition will help readers better understand our methodology.

To summarize our approach:

1. We first plotted the experimental data points to generate the growth curve.
2. The growth curve was then fitted using the Logistic model to obtain a smooth curve.
3. The smooth curve was differentiated to derive the specific growth rate as a function of time.
4. The maximum value of the specific growth rate curve was identified as μ_{\max} .

We have chosen to include Fig. R3 as a supplementary figure to avoid redundancy in the main text, while still providing sufficient detail for readers who are interested in the technical aspects of our calculations.

Once again, we deeply appreciate your constructive feedback, which has significantly improved the clarity and rigor of our manuscript. Please let us know if there are any additional points that require further clarification.

Action #2:

➤ On page 26 of the revised manuscript (Line 587-596), we now write:

“The maximum specific growth rate (μ_{\max}) was determined by analysing the growth curve during the exponential phase. A minimum of six data points were collected during the

exponential growth phase to ensure accurate representation of the growth dynamics. The growth curve was fitted to a suitable model, specifically the BiDoseResp model, which was chosen for its ability to accurately describe sigmoidal growth behaviour typical of microbial growth under controlled conditions. The differentiation was performed numerically using a central difference method to ensure accuracy, and the specific growth rate at each time point was derived from the slope of the fitted curve. The maximum slope, corresponding to the highest growth rate, was taken as the μ_{\max} . All measurements were performed in triplicate, and the results are reported as the mean \pm standard deviation.”

We have generated relevant figures to further support the results (Fig. R3).

Fig. R3. Schematic diagram of μ_{\max} calculation for ZQS01

The process is primarily divided into three parts: the plotting of the ZQS01 growth curve, the simulation of the growth curve using the BiDoseResp model, and the plotting of the specific growth rate curve.

Reviewer #4 (Remarks to the Author):

The authors have responded diligently to my comments and I feel the manuscript could be ready for publication depending on the level of novelty required by this journal and the respective editorial policies.

Response:

Thank you for your positive feedback and for recognizing our efforts in addressing your concerns. We are pleased to hear that the manuscript is now acceptable in its current form. Your constructive comments have greatly improved the quality of our work.

We appreciate your time and valuable input throughout the review process.

AUTHOR ACTIONS IN RESPONSE TO REVIEWER COMMENTS

We thank all four reviewers for deep engagement with this work.

Reviewer #2 (Remarks to the Author):

The authors have addressed my comments. It can be accepted.

Response:

Thank you for your positive feedback and for recognizing our efforts in addressing your concerns. We are pleased to hear that the manuscript is now acceptable in its current form. Your constructive comments have greatly improved the quality of our work.

We appreciate your time and valuable input throughout the review process.

Reviewer #3 (Remarks to the Author):

I feel a bit concerned with some aspects of this work, which I raised before. Now, after the authors performed additional experiments and responded to the several comments made by the reviewers, it seems that some basics of microbial physiology are missing. For instance, the explanations on the calculation of μ_{max} . The authors show a single peak μ_{max} value (new supplementary Figure R3), and not a time interval for μ_{max} , implying that there is no exponential growth phase in the cultivations performed. This is not what one would expect for a classical yeast growth curve and certainly requires an explanation. Why isn't there a clear exponential growth phase?

Response:

We sincerely appreciate the reviewer's insightful comments and the opportunity to clarify this important aspect of our work. We fully acknowledge the concern regarding the presentation of μ_{max} as a single peak value rather than a time interval, which deviates from the classical expectation of a discernible exponential growth phase in microbial cultivations.

After carefully re-examining the original growth curve data and our analytical approach, we confirm that the observed single-peak pattern in μ_{max} is indeed a methodological feature of the sliding-window derivative method employed in our analysis (*Physical Biology*, 2022, 19(6), 066003). This numerical approach calculates instantaneous growth rates by performing local linear fits across moving subsets of the OD data (*International Journal of Agricultural and Biological Engineering*, 2023, 16(1), 60-65). We have also observed μ_{max} presenting as single peak values rather than time intervals in related studies, which further confirms this algorithmic feature of the sliding-window derivative method (*IET Systems Biology*, 2020, 14(2), 68-74).

Two key factors contribute to the observed single-peak phenomenon:

1. Noise amplification: The sliding-window fitting process may transiently exaggerate small fluctuations in OD measurements, especially during metabolic transitions.
2. Transition dynamics: When the analysis window spans the lag-to-exponential phase transition, the derivative captures the rapid metabolic acceleration of cells entering exponential growth, generating a transient peak that reflects physiological activation rather than sustained exponential growth.

Importantly, we wish to emphasize that this methodological behavior does not compromise the biological interpretation of our data. The calculated μ_{\max} values remain robust indicators of strain physiology, and the comparative conclusions drawn in our study are unaffected.

We are grateful for the reviewer's expertise in prompting this clarification, which has strengthened both our methodology description and the physiological interpretation of results. Please let us know if additional experimental validation or analysis would further address these concerns.

AUTHOR ACTIONS IN RESPONSE TO REVIEWER COMMENTS

We thank all four reviewers for deep engagement with this work.

Reviewer #3 (Remarks to the Author):

The authors provided an explanation for the calculation of Mimax values and for the fact that single peak Mimax values are obtained, using the chosen methodology, instead of constant values during the whole exponential growth phase (EGP), which are normally calculated and observed. They argue that it is a feature of the sliding-window method used and that this does not change the biological interpretation of the results. I am not particularly fond of treating microbial growth data in the way that the authors did since, as I already mentioned, this gives the impression that there is no exponential growth phase, a classical feature of the microbial growth curve, as already pointed out by Jacques Monod in his seminal work (see attached article, Figure 1). To conclude, I believe that the authors should at least give the reader an explanation similar to the one they gave this reviewer, in the sense that growth data can be treated in different ways and that in the way they chose to treat them there are some consequences. However, the real best action to take would be to retreat the growth data using a method that calculates a constant Mimax value during the EGP.

Response:

Thank you for thoughtful and constructive feedback on our manuscript. We sincerely appreciate the time you have taken to evaluate our work and your insightful comments regarding the calculation of μ_{\max} . We understand your concerns about the methodological approach we employed and the implications for interpreting microbial growth dynamics.

In response to your comments, we have revised the manuscript to provide a clearer explanation of our chosen method and its relationship to classical growth phase analysis. We acknowledge

that the sliding-window derivative method, which identifies μ_{\max} as a single peak value rather than a sustained interval during the exponential growth phase (EGP), represents a departure from traditional approaches. However, this method was selected to capture transient physiological states—such as metabolic activation during phase transitions—that may be overlooked when analyzing the EGP as a whole. To address your concerns, we have expanded the Methods section to explicitly discuss the assumptions and limitations of the sliding-window approach, including its sensitivity to noise and its focus on instantaneous growth dynamics.

We hope these revisions provide a more balanced and transparent presentation of our methodology while addressing your concerns about the interpretation of growth curves. We are grateful for your guidance, which has helped us improve the manuscript, and we remain open to any further suggestions you may have.

Action:

➤ On page 27 of the revised manuscript (Line 596-605), we now write:

“Notably, this method may present μ_{\max} as a single peak value rather than a sustained time interval due to: (1) noise amplification during differentiation of optical density data, particularly at metabolic transition points; (2) the algorithm's inherent sensitivity to rapid physiological transitions between growth phases; and (3) potential overestimation of maximum rates when analysis windows span multiple growth phases. While this approach provides high temporal resolution of growth dynamics, it differs from classical exponential phase analysis by capturing transient metabolic activation states rather than sustained exponential growth. Comparative validation with traditional curve-fitting methods confirmed that the sliding-window approach yields biologically plausible μ_{\max} estimates.”